# Local flux coordination and global gene expression regulation in metabolic modeling

Gaoyang Li[1,2], Li Liu[3], Wei Du [1] ✉ & Huansheng Cao [3] ✉

Genome-scale metabolic networks (GSMs) are fundamental systems biology representations of a cell's entire set of stoichiometrically balanced reactions. However, such static GSMs do not incorporate the functional organization of metabolic genes and their dynamic regulation (e.g., operons and regulons). Specifically, there are numerous topologically coupled local reactions through which fluxes are coordinated; the global growth state often dynamically regulates many gene expression of metabolic reactions via global transcription factor regulators. Here, we develop a GSM reconstruction method, Decrem, by integrating locally coupled reactions and global transcriptional regulation of metabolism by cell state. Decrem produces predictions of flux and growth rates, which are highly correlated with those experimentally measured in both wild-type and mutants of three model microorganisms *Escherichia coli*, *Saccharomyces cerevisiae*, and *Bacillus subtilis* under various conditions. More importantly, Decrem can also explain the observed growth rates by capturing the experimentally measured flux changes between wild-types and mutants. Overall, by identifying and incorporating locally organized and regulated functional modules into GSMs, Decrem achieves accurate predictions of phenotypes and has broad applications in bioengineering, synthetic biology, and microbial pathology.

Cellular life maintains itself and replicates through the entire set of biochemical reactions in genome-scale metabolic networks (GSMs) operating in a well-coordinated manner[1–3]. Such a resilience is achieved through complex coordination on a systems scale, mainly determined by local and global regulation[4,5]. First, the microbial flux state appears to evolve toward optimality under one growth condition and a minimal adjustment between (environmental or genetic) conditions[6]; consequently, local metabolite levels are stable in the face of environmental or genetic perturbations[5–11] due to quick compensation from local reactions. This suggests that local rerouting of fluxes in GSMs operates efficiently, which plays a crucial role in metabolite homeostasis in maintaining global optima under one condition or across conditions[6]. Such quick compensations come as a result of the evolution of

metabolic networks, in which reactions and metabolites preferably attach to high-efficiency biochemical reaction chains in an organism, e.g., pathways or biological processes, according to the principles of network growth and preferential attachment[12–14]. This mode of network organization leads to a topologically local coupling for metabolic flux rerouting. For example, in the TCA (tricarboxylic acid) cycle, the reaction chain D-isocitrate --> a-ketoglutarate -->... --> malate ($K_m = 0.029\,\mu M$ and $K_{cat} = 106.4$ for IDH3 (Isocitrate dehydrogenase)) is preferred as the primary branch for oxidation of acetyl-CoA over the low-efficiency reaction chain D-isocitrate --> succinate ($K_m = 8\,\mu M$ and $K_{cat} = 28.5$ for AceA (Isocitrate lyase)). Recent discoveries in both network properties and enzyme parameters, e.g., $K_m$ and $K_{cat}$, support such network organization[15,16]. Some reactions are more closely

[1]Key Laboratory of Symbolic Computation and Knowledge Engineering of Ministry of Education, College of Computer Science and Technology, Jilin University, Changchun 130012, China. [2]Translational Medical Center for Stem Cell Therapy and Institute for Regenerative Medicine, Shanghai East Hospital, Bioinformatics Department, School of Life Sciences and Technology, Tongji University, Shanghai, China. [3]Division of Natural and Applied Sciences, Duke Kunshan University, Kunshan 215316, China. ✉e-mail: weidu@jlu.edu.cn; hc284@duke.edu

interconnected with high kinetic capabilities than others and form "small world" structures in GSMs[17–19], particularly in central metabolism[20]. Therefore, those topologically coupled reactions constitute the biochemical properties-derived local metabolic autoregulation and flux coordination in GSMs. The local metabolic coordination-guided regulation is manifested in three aspects. First, the changes due to local internal perturbations (gene deletions) are quickly compensated by neighboring reactions[5]. Second, functionally related genes are organized as operons coregulated in microorganisms[21–25]. Third, failure to recognize the coupled reactions leads to poor performance in perturbation effect estimation by steady-state linear flux optimization, e.g., flux balance analysis (FBA).

Besides local fast-acting flux adjustment, global metabolic homeostasis is achieved through transcriptional regulation[9,26–30], which accounts for the majority (about 70%) of the changes in gene expression between growth conditions[29]. The gene expression profiles are primarily regulated by the global growth state via the sequestration or release of transcription factors (TFs) with the variation in the concentration of growth indicator metabolites[28,29], as shown in the activities of over 200 TFs showing strong correlations with few cognate metabolites following the transition from starvation to growth in *E. coli*[9]. For example, cyclic AMP, fructose-1,6-bisphosphate, and fructose-1-phosphate bind to TFs (e.g., *Crp* and *Cra*) and thereby mediate most of the specific transcriptional regulation. However, there is a low correlation between biochemical reaction rate and its enzyme expression level. For example, in central metabolism, only a few enzymes change proportionally to flux changes in the reactions in the tricarboxylic acid cycle (TCA Cycle)[4]. Consequently, the triangular regulatory relationship of metabolites->TFs->transcription cannot sufficiently reflect the reaction flux variation[4,31–33], and this insufficiency has hampered the integration of transcriptional regulation with the current stoichiometric matrix-based GSM models[4,31]. Few studies attempting to integrate the kinetic parameters are Michaelis-Menten equation-based genome-scale multi-omics data fitting[34,35], but do not consider the global transcriptional regulation of metabolism. In practice, it is challenging and expensive to obtain complete enzyme kinetic parameters (i.e., $K_m$ and $K_{cat}$) from paired metabolomics and proteomics data to build a reaction kinetics-constrained metabolic flux prediction model in GSMs.

In this work, we present a GSM model, Decrem, to quantitatively characterize the local topological cooperation regulation and the global transcriptional regulation. For this purpose, we integrate local flux coordination and transcriptional regulation of global growth state-mediated key metabolic reactions into Decrem to approximate flux distribution (Fig. 1). We first derive a decoupled Decrem model by analyzing the cooperated topological profiles of GSMs and incorporating them into the canonical FBA by representing the synchronously coordinated (coregulated) and closely coupled reactions with a group of independent sparse bases (reactions) according to a stoichiometric matrix decomposition. We test Decrem in three model organisms: *E. coli*, *S. cerevisiae*, and *B. subtilis*. The flux distributions predicted using Decrem are highly consistent with experimentally measured fluxes in multiple strains (wildtype and mutants). Then, the growth state-regulated fundamental enzyme kinetics are identified to create kinetic Decrem to model the global dynamic transcriptional regulation of metabolic networks in response to environmental perturbations. Unlike the previous kinetic models, which often focus each essential metabolite/flux on its corresponding enzyme kinetics, we focus our attention on the enzymes directly regulated by growth state (biomass composition) because they represent the more significant transcriptional regulation than the other reactions. A specific advantage of the kinetic Decrem model is that only several growth state-related metabolites suffice to achieve transcriptional regulation and thus reduce the requirements for necessary kinetic parameters and paired multi-omics data. The accurate growth rates predicted by the kinetic Decrem

model in *E. coli* genome-scale knockout strains revealed that intracellular perturbations are mainly 'buffered' by highly coupled reactions, which reveals the coordination between crucial precursors of central metabolism and cell growth. Overall, we recognize metabolic regulation as the local topological coordination and global growth-related key transcriptional regulation, which demonstrates that Decrem can integrate metabolic regulation into current GSM models.

## Results

### Reconstruction of GSMs with topologically decoupled reactions

Through comprehensive multi-omics data analysis, we find that the transcriptional data of metabolic genes have no significant correlation with the corresponding $^{13}C$ isotope fluxes in *E. coli* central metabolism (Fig. 2a). In contrast, a high correlation is observed between the fluxes of local topologically coupled neighboring reactions, e.g., the element reactions of conventional pathways in central metabolism: the average correlation coefficient $r$ is 0.913, 0.975, and 0.794 for the reactions in glycolysis, PPP (pentose phosphate pathway) and TCA (tricarboxylic acid) cycle, respectively, as opposed to 0.505, 0.267, and 0.421 for each uncoupled reaction set (t-test, p = 2.33E−9, 3.06E−43, and 2.25E−4, respectively; Fig. 2b), as well as the metabolic gene expression of those local neighboring reactions (Fig. 2c). These correlations suggest potential coordinated regulation of locally coupled neighboring reactions. Here, we develop a topologically decoupled linear representation of the metabolic network to characterize the coactivated regulation of topologically highly coupled reactions with three steps. First, substructures composed of tightly connected local reactions in the metabolic network are identified from its bipartite graph representation (Supplementary Fig. 1)[36], with a topological coupling metric as the number of simple cycles between two reaction nodes in the bipartite graph. Specifically, the identified coupled reaction subnetwork included 927 of the 2382 reactions in the *E. coli* model *i*AF1260 (Supplementary Data 1). In central metabolism, such as glycolysis, PPP, TCA cycle, amino acid, and glycerophospholipid pathways, the reactions especially the reversible reactions (70%, 141 of 201) primarily consist of coupled reactions. In contrast, tRNA, membrane lipid biosynthesis, membrane transport pathways, and the biomass reaction primarily consist of uncoupled linear reaction chains (Fig. 2d). The $K_m$ values (0.023 mM) of identified coupled reactions are smaller (by 56.5%) than those (0.036 mM) in the uncoupled reaction chains (Wilcoxon test, $p = 5.74E{-}4$; Fig. 2e). Together, the high correlation of gene expression and fluxes among local topologically coupled reactions, the biochemical proximity, and high substrate affinity of enzyme catalysis suggest that these coupled reactions prefer to locally cooperate and be co-regulated to quickly respond to environmental perturbations, especially in central metabolism, before reaching out to more distant reactions. Next, we decompose the highly coupled reaction substructures into their linear representations with minimal independent reaction components, using sparse linear basis (SLB) vectors of null space of their corresponding stoichiometry matrix. Each SLB consists of the least number of coupled reactions to form an indivisible independent flux (see Methods). Like elementary flux modes (EFMs)[37], metabolic fluxes could be decomposed as the weighted linear combination of the identified SLBs[10]. However, unlike the almost infinite number of EFMs for large GSMs, there is a unique number of SLBs to define the mutually independent components for the densely coupled reactions[38]. We validate the coordinated activation of reactions within the SLBs through gene expression in 24 knockout strains of *E. coli*[38]. Indeed, the correlations (mean $r = 0.447$) of gene levels among the element reactions from the same SLBs are higher than those (mean $r = 0.28$) from different SLBs in central metabolism (t-test, $p = 2.05E{-}33$; Fig. 2f). This suggests that the reactions from the same SLBs tended to be coactivated, but the reactions in different SLBs are more independent than those from the same SLBs. To explore the local coregulation of SLBs, we enrich the constituent genes of SLBs into all the TFs of *E.*

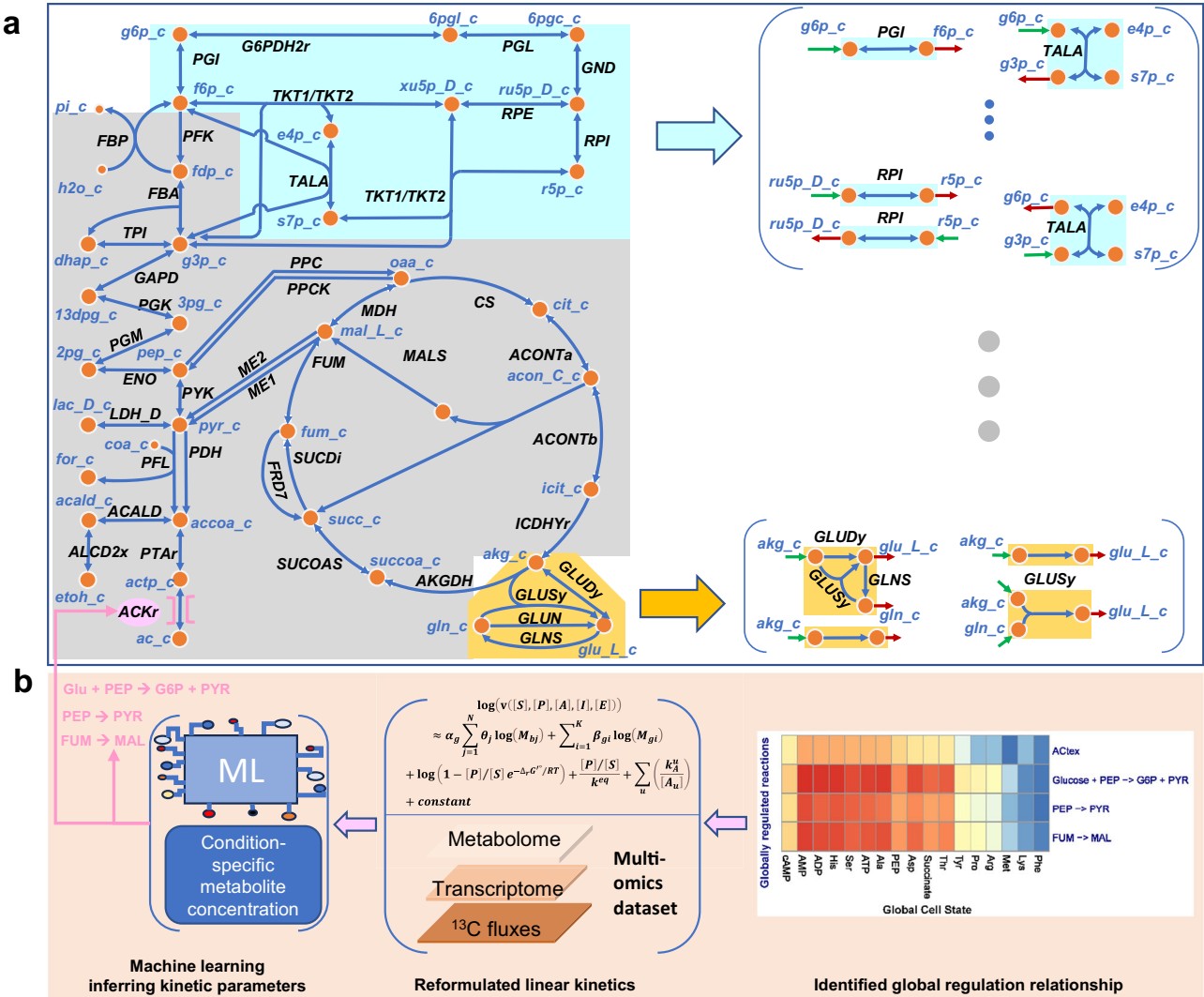

**Fig. 1 | The schematic diagram of the Decrem model. a** Decrem represents the local coupled and coordinated reactions as a group of SLBs to constrain the flux distribution. Two examples of coupled reactions are provided here in cyan and brown background (left side of Panel **a**); each coupled reaction set is decomposed into SLBs (right side of Panel **a**). **b** A reformulated linear kinetics is built for the identified globally regulated reactions and is used to bind those globally regulated fluxes. The middle part in the blue box predicts condition-specific fluxes using the reformulated linear kinetic, and the left blue box infers kinetic parameters. ML machine learning.

*coli* and find a significant enrichment TF set for SLBs (Fig. 2g). Most of the enriched TFs are part of the general DNA-binding transcriptional regulators, such as SoxS and NrdR. These regulations are often determined by cell metabolic state (superoxide or nitric oxide and ATP concentration).

In the last step, we reconstruct a GSM model, Decrem, by merging the element reactions of each SLB into a linear basis reaction (LBR) with reallocated stoichiometric coefficients (see Methods). To explore the variable range of fluxes in Decrem against the original GSMs, flux variability analysis (FVA) is conducted in *E. coli* and *S. cerevisiae*. The results confirm the preservation of solution space (Supplementary Fig. 2)[39]. Our model reassigns the flux ranges, which vary with different pathways, and makes the distribution of flux variability in central metabolism more consistent with the experimental [13]C flux distribution, which may reduce the uncertainty of original GSMs, and serves as our working model for subsequent functional analyses.

### Benchmark the metabolic fluxes prediction in response to environmental perturbation in vivo/vitro
We first apply Decrem for metabolic flux prediction with an FBA strategy (see Methods), in comparison with three other methods

which are carried on the canonical GSMs: FBA, pFBA (parsimonious FBA), and RELATCH[40–42] in model organisms of *E. coli* (*i*AF1260 and *i*ML1515), *S. cerevisiae* (*i*MM904), and *B. subtilis* (*i*YO844). These four models contain 766, 1104, 558, and 332 LBRs in the corresponding decoupled Decrem models, respectively (Supplementary Data 1).

We calculate the metabolic flux distribution in the canonical *E. coli* *i*AF1260 model under MOPS medium supplemented with glucose or xylose and under aerobic or anaerobic respiration[43] (Supplementary Note 1), with nutrients as the sole constraints on GSMs (Supplementary Data 2). The predictions using Decrem have higher correlations with the experimentally measured [13]C-MFA (metabolic flux analysis) fluxes (Supplementary Data 3) than the predictions by three other methods on the original GSMs under all four conditions, in all three metrics (Fig. 3a, c, d). Meanwhile, Decrem flux predictions have the smallest MSE (maximum upper bound of flux being 1000 mM/gDW/hr), and most of the activated reactions (with nonzero flux) agree well with the experimental [13]C-MFA fluxes (Fig. 3c, d). Here, we provide an example of the superior performance of Decrem in the TCA cycle. The Decrem predictions are consistent with the experimentally measured [13]C-MFA fluxes (Supplementary Fig. 3), whereas four activated reactions are predicted as inactive (zero fluxes) by FBA. Given the multiple versions

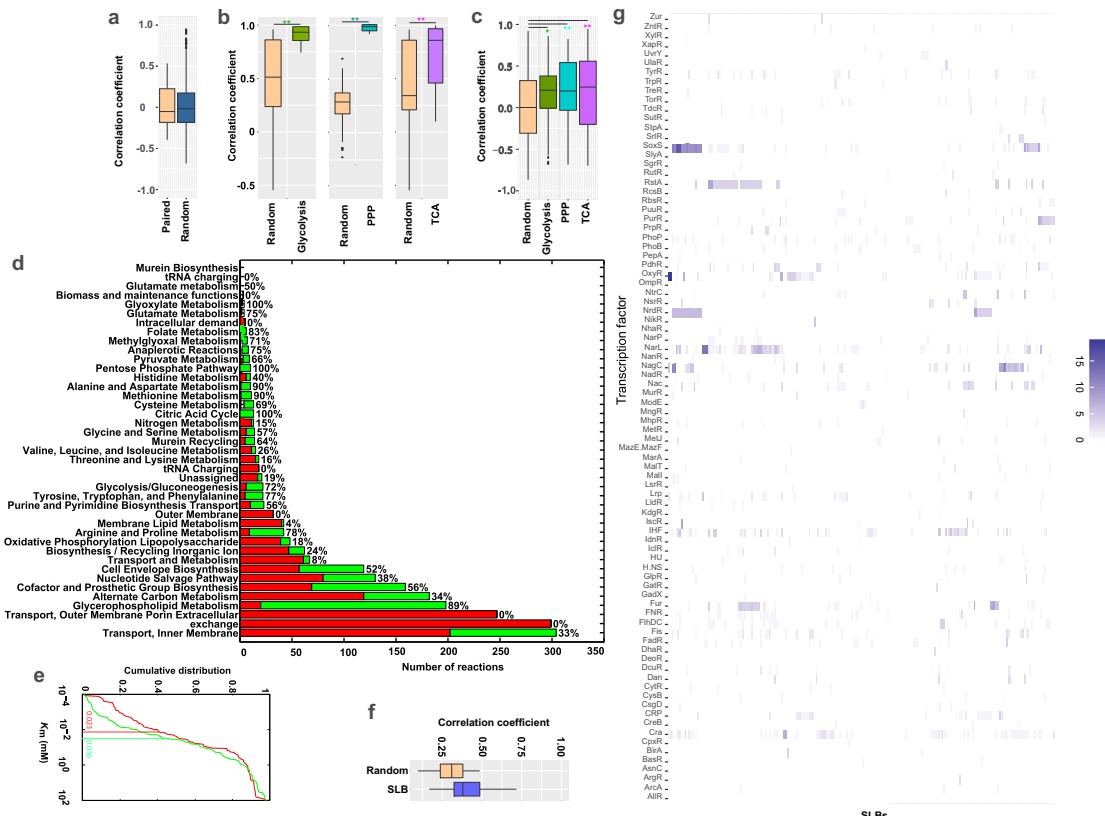

**Fig. 2 | The statistical analysis of decomposed topological components of *E. coli* iAF1260 model. a** The correlation distribution between paired [13]C MFA flux and gene expression ($n = 37$) vs the correlation between random gene and flux ($n = 1394$) in 24 strains (unpaired two-tailed *t*-test, $p > 0.05$). **b** The correlation distribution [13]C MFA fluxes within each of three pathways of central metabolism ($n = 15$ for glycolysis, $n = 21$ for PPP, and $n = 21$ for TCA), compared to the corresponding non-coupled reactions (from different pathways) ($n = 93$ for glycolysis, $n = 78$ for PPP, and $n = 78$ for TCA) (unpaired two-tailed *t*-test, *$p < 0.05$, **$p < 0.01$). **c** the correlation distribution gene expression within each of three pathways ($n = 171$ for glycolysis, 66 for PPP, and 190 for TCA pathway) of central metabolism, compared to the corresponding random ($n = 3570$) (unpaired two-tailed *t*-test, *$p < 0.05$, **$p < 0.01$). **d** The distribution of the proportion of coupled reactions (green bars) and non-coupled reactions (red bars) among the 41 metabolic

pathways. **e** The comparison of the accumulated distribution of kinetic constant $K_m$ between the coupled reactions (red line, $n = 153$) and non-coupled reactions (green line, $n = 161$), the vertical bars are the mean $K_m$ values of the two groups (Wilcoxon rank-sum test, $p = 5.74E−4$). **f** The correlation distribution of gene expression among the element reactions of sparse linear basis (SLB, $n = 266$) vectors against the gene expression among the non-coupled reactions ($n = 443$) (unpaired two-tailed *t*-test, $p = 2.05E−33$). **g** The heatmap shows that most identified SLBs are significantly enriched for several general DNA-binding TFs. In **a**–**c** and **f**, the black center line denotes the median value (50th percentile), while the boxes of various colors contain the 25th to 75th percentiles of the dataset. The black whiskers mark the 5th and 95th percentiles, and values beyond these upper and lower bounds are considered outliers, marked with black dots. Source data are provided as a Source Data file.

of available *E. coli* GSMs with different numbers of reactions and distinct completeness, we test the generality of Decrem on six *E. coli* K-12 BW25113 knockout mutant strains with the newest *E. coli* iMF1515 model[11] (Supplementary Data 2 and Data 3), and find a consistent higher correlation (higher activated reaction numbers and lower MSE) of Decrem than all three other methods (Fig. 3b–d), which indicates the properties of local reaction coordination can effectively improve the flux prediction of current GSMs.

Decrem also outperforms the other three methods in flux prediction in *S. cerevisiae* and *B. subtilis*[44,45] (Fig. 3a, c, d; Supplementary Note 1, Supplementary Data 1–3). Particularly, Decrem FBA, pFBA, and RELATCH tested with conventional complex eukaryote *S. cerevisiae* model produce *r* to be 0.696, 0.3, 0.284, and 0.3, respectively, in which reactions are often strongly coupled based on cellular compartments. To validate the flux distribution in the mitochondrial compartment, we build a reference flux distribution for the *S. cerevisiae* iMM904 model using the wildtype [13]C-MFA fluxes (Supplementary Note 1), and then compared the reference fluxes with the predicted reaction fluxes of mitochondrial reactions from Decrem and original GSMs. The resulting number of co-occurring nonzero fluxes are 76 and 50, and the Spearman correlation coefficients are 0.674 and 0.462, respectively.

We notice the specific nonzero flux reactions by Decrem are related to oxidative phosphorylation and transportation, such as proline oxidase NAD (Supplementary Data 2). To further explore whether Decrem can predict the flux range varying across the diverse perturbation, FVA is carried out on Decrem, the original *E. coli* iML1515, and *S. cerevisiae* iMM904, respectively. We find a higher Jaccard index metric between the Decrem predictions and [13]C-MFA-estimated 95% confidence intervals across various mutant strains, compared with predictions using the original GSMs (Fig. 3e and Supplementary Fig. 4; Supplementary Data 3–4; Supplementary Note 1). This improved prediction indicates that Decrem reassigns the distribution of flux variability, which may reduce the uncertainty of original GSMs.

### Decrem accurately identifies the mutant fluctuation in *Yeast* knockout strains
We first evaluate Decrem in predicting fluxes in response to genetic perturbation (single-gene deletions) in two GSMs of *S. cerevisiae*, iDN750 (1059 metabolites and 1266 reactions) and iMM904 (1226 metabolites and 1577 reactions). Thirty-eight mutants with experimental [13]C-MFA fluxes, growth rates, nutrient uptake properties, and several extracellular exchange fluxes[46] are used (Supplementary

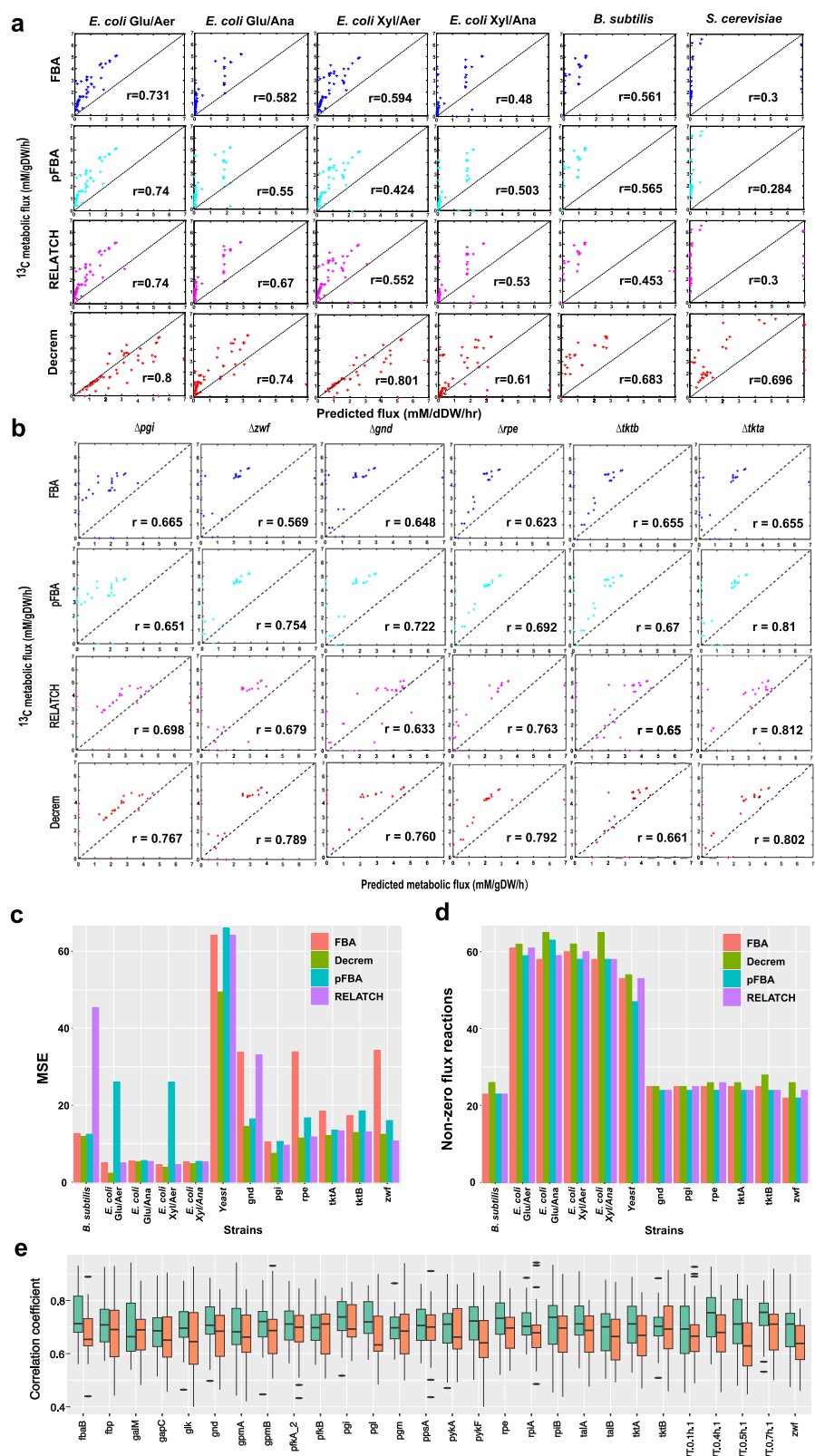

Note 2; Supplementary Data 4). Compared to four other methods, pFBA, FBA, RELATCH, and REPPS[47]. The Decrem flux predictions show the highest correlations with the experimentally measured fluxes in almost all mutant strains for both GSMs (Fig. 4a and Supplementary Data 5). Specifically, the average Spearman and Pearson correlation coefficients (*r*) determined by Decrem for *i*DN750 are 0.72 and 0.763 in the 38 mutant strains, compared to 0.615 and 0.733, 0.607 and 0.681,

0.556 and 0.707, 0.462 and 0.595 for pFBA, FBA, RELATCH, and REPPS, respectively (one-way ANOVA, *p* = 5.52E−20 and 4.21E−07). For the complete *Yeast i*MM904, the mean Spearman and Pearson's correlation coefficients are 0.782 and 0.936, 0.765 and 0.747, 0.656 and 0.915, 0.75 and 0.925, 0.645 and 0.841 for Decrem, pFBA, FBA, RELATCH, and REPPS, respectively, which are significantly different (one-way ANOVA; *p* = 4.35E−06 and 8.26E−06, respectively). The differential accuracy of

**Fig. 3 | Comparison of the predictions in three model microorganisms by FBA, pFBA, RELATCH, and Decrem. a** Metabolic flux distribution between the experimental measurements and predictions by the four methods using *E. coli* *i*AF1260 under four conditions: glucose (Gly) or xylose (Oxy) under aerobic (Aer) or anaerobic (Ana) respiration, as well as in the *S. cerevisiae* *i*MM904 model and *B. subtilis* *i*YO844 model with glucose under aerobic respiration. *r* is the Pearson correlation coefficient between the measured $^{13}$C MFA fluxes and the predicted fluxes. **b** Metabolic flux distribution between the experimental measurements and predictions by the four methods in *E. coli* *i*ML1515 with *pgi*, *zwf*, *gnd*, *rpe*, *tkta*, and *tktb* genes knockout mutant strains. **c** Comparison of MSE among the four methods across the 12 growth conditions. **d** Comparison of nonzero flux reactions among the four methods across the 12 growth conditions. **e** Comparison of flux variability range consistency between Decrem predicted and $^{13}$C MFA fluxes across 28 mutant strains (green) and the prediction by original *i*ML1515 model (red) (*n* = 33 samples for each subplot, paired two-tailed *t*-test, *p* < 0.01). The correlation coefficient is measured by the Jaccard index. The black center line denotes the median value (50th percentile), while the green and red boxes contain the 25th to 75th percentiles of the dataset. The black whiskers mark the 5th and 95th percentiles, and values beyond these upper and lower bounds are considered outliers, marked with black bars. *n* = 31 samples are included in each box. Source data are provided as a Source Data file.

flux predictions between the *i*DN750 and *i*MM904 models shows that Decrem performed better than other methods, particularly on metabolic models that are incomplete (e.g., *i*DN750 vs. *i*MM904) (Fig. 4a). Further analysis revealed that such differences exist because a large proportion (47%, 146 of 311) of the reactions in *i*MM904 (vs. *i*DN750) are highly coupled reactions, compared to an average of 35% (558 of 1577) for *i*DN750 (Fisher's exact test, (*k* = 146, *m* = 558, *n* = 311, *N* = 1577 for hypergeometric distribution) *p* = 5.427e−04); this increased reaction coupling may optimize the solution space of the metabolic model and change the optimal reaction paths in *i*MM904. In addition, we explore the prediction difference of mutant strains across specific pathways by MSE for all the methods used and find that Decrem presents a significantly smaller MSE than other methods (Fig.4b and Supplementary Fig. 5), especially for mutants in central metabolism, which contain many coupled reactions, e.g., *FUM1*, *MDH1*, and *PDA1* mutants in TCA cycle and pyruvate metabolism, respectively.

We then apply Decrem and pDecrem (Decrem model with parsimonious FBA optimizer, see Methods) to estimate the growth rate in the 38 mutant strains of *S. cerevisiae* using the *i*MM904 model. PCA of the predicted fluxes among the mutant strains from all six methods shows that the top two principal components (PCs) can explain more than 99% of the variance of all predicted fluxes. However, the two top PCs predicted by Decrem are each highly correlated (*r* > 0.7) with experimentally measured growth rates. At the same time, the other methods have, at most, one PC that merely shows moderate correlation (*r* ~ 0.5) (The PCs correlation *r* > 0.8 from $^{13}$C MFA fluxes; Supplementary Data 5). Furthermore, a PCA regression reveals that Decrem prediction has the best flux variance to explain the observed growth rates, showing the highest coefficient of determination ($R^2$): 0.9 (Decrem) and 0.9 (pDecrem) vs. 0.74, 0.731, 0.731, 0.841, 0.9, and 0.9 for the other methods (Supplementary Fig. 6). Importantly, Decrem can both correctly identify and explain the six mutations with significant growth effects reducing growth rate to <0.5 h$^{−1}$: *ALD6*, *FUM1*, *PDA1*, *RPE1*, *MDH1*, and *ZWF1* (Fig. 4c), whereas the other methods can only identify some of them. Strikingly, Decrem-predicted flux distribution through the specific pathways correctly explains the significant fluxes rewiring in these 'exceptional' mutants as experimentally observed[46,48]; none of the other four methods captured these flux responses with the original GSMs (Fig. 4d–f). Specifically, Decrem fluxes accurately classify the two groups of redox metabolic fluxes: NADP$^+$/NADPH-related mutants: (*ZWF1*, *RPE1*, and *ALD6*), and NAD$^+$-related mutants in TCA: (*FUM1*, *PDA1*, and *MDH1*)[43] (Fig. 4d). Interestingly, Decrem also distinguishes the *ZWF* mutant from the ALD6 and RPE1 mutants, as having increased mitochondrial fluxes (Fig. 4d), precisely as experimentally observed[46,48]. That is, the exceptionally high fluxes of the mitochondrial transport pathway and TCA pathway in the *ZWF1* strain agree with the experimental observation that NADPH and NADP$^+$-dependent mitochondrial malic enzyme flux is significantly increased (Fig. 4g)[46]. The other methods identified only the mutations with large growth effects, leading to a high incorrect rate. In comparison, Decrem demonstrates high accuracy and low false positive rates in assessing the growth rate and well approximates the real metabolic fluxes in the mutants.

## Integrating global transcriptional regulation-derived key reaction kinetics into Decrem

Cellular metabolism is rather dynamic but transcriptional regulation is insufficient to explain flux change[4,35], which presents a major obstacle to the multi-omics integration of metabolism. We present here that the biomass/growth rate, rather than regulator metabolites, plays a dominant role in the activity for most of gene expression in central metabolism. This insight provides a practical strategy to quantify the cooperation relationship between the biomass/growth state-regulated metabolic genes and their kinetic flux.

For this purpose, we first investigate the correlation between potential regulator metabolites obtained in this study through an extensive literature and database survey[49,50] and metabolic gene expression in central metabolism on a multi-omics dataset of 24 single-gene knockout strains of *E. coli*[49]. They include the expression of 85 metabolic genes in central metabolism, the concentrations of over 100 metabolites and 51 $^{13}$C-MFA fluxes for each strain[49]. In total, 45 selected regulator metabolites (and biomass constituents) are classified into two groups according to their concentrations: the biomass-constituent group (BG) and the precursor or regulator metabolite group (PG) (Fig. 5a and Supplementary Data 6). Most BG metabolites have a high positive correlation with the genes in PPP and pyruvate metabolism and have the negative correlation with some genes of the TCA cycle. At the same time, the PG only presents a few coregulated metabolites, e.g., G6P, F6P, and AMP et al. (Fig. 5a). This difference suggests the dominant role of biomass/growth rate in transcriptional regulation. To validate the observation, we obtained experimental metabolite and gene expression data by growing *E. coli* BW25113 on MOPS minimal medium on a time series. Indeed, a similar correlation is observed (Supplementary Data 6 and Supplementary Fig. 7).

We then develop a linear transcriptional regulation mechanism to explain the observed correlation (see Methods) and validate this mechanism by conducting a partial least squares regression (PLSR) to quantitatively fit the linear global regulation of the observed expression profiles of the 85 genes to the concentrations of the potential regulatory metabolites of either group (see Methods). By taking stringent combined thresholds (total regression correlation *r* > 0.84 and the correlation of first PC > 0.38 according to PLSR; Supplementary Note 3), 32 of the 85 genes are identified as the regulatory targets of the 23 BG metabolites, whereas no genes are identified as the regulatory targets of the PG metabolites (Fig. 5b and Supplementary Fig. 8). The identified metabolite-gene regulatory pairs are verified by the high correlations between the identified 32 globally regulated genes and BG metabolites using canonical correlation analysis on our experimental dataset, against the poor canonical correlation of PG metabolites with all studied genes (all genes vs. all metabolites and all genes vs. BG) (Fig. 5c, d and Supplementary Fig. 7b,c). We further test the statistical significance of identified correlations between the measured and predicted gene levels using a wide range of metabolites selected by 10,000 random samplings from the 45 potential metabolic regulators (Supplementary Note 3). A *p*-value of 3.1E−3 for the 23 identified growth-associated metabolites is observed against the randomly selected metabolites (Fig. 5e), while the *p*-value is 0.48 for the

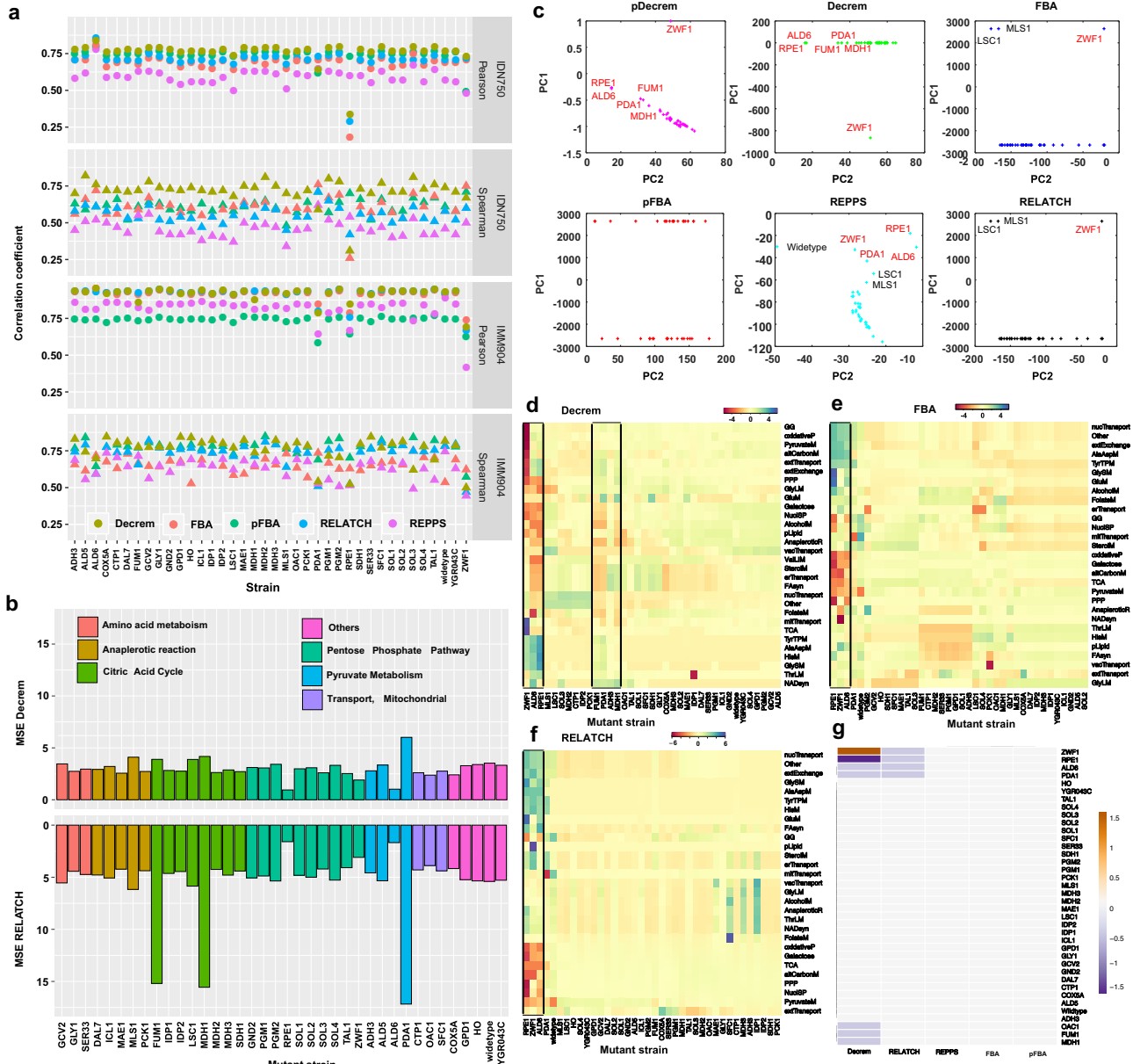

**Fig. 4 | Comparison of the flux predictions in two *S. cerevisiae* metabolic models. a** The Pearson and Spearman correlation between the $^{13}$C MFA measured and predicted metabolic fluxes across the five test methods (FBA, pFBA, RELATCH, REPPS, and Decrem) among 38 strains in the *i*MM904 and *i*DN750 models. ($n = 38$, group = 4, one-way ANOVA test for each subplot, $p = 4.35E-06$ for Pearson correlation and 8.26E−06 for Spearman correlation for *i*MM904 model respectively, and $p = 5.52E-20$ for Pearson correlation and 4.21E−07 for Spearman correlation for *i*DN750 model, respectively). **b** Comparison of MSE among the pathways in central metabolism by Decrem and Relatch across 38 growth conditions. **c** The top two principal components of predicted flux variances among the 38 mutant strains, which were predicted from the six test methods. The name of deleted genes with high adverse effects on growth are colored red and the genes without significant effects on growth are colored black. **d**–**f** The pathway-specific fluxes distribution across 38 mutant strains predicted by Decrem (**d**), FBA (**e**), and RELATCH (**f**), the boxes indicate the mutant strains from the same pathways share a similar flux distribution pattern. **g** Decrem predicted fluxes on the mitochondrial transport pathway and TCA for *ZWF* mutant strains vs. the RELATCH, REPPS, FBA, and pFBA. Source data are provided as a Source Data file.

PG metabolites. In addition, we validate the identified 32 genes regulated by global BG metabolites using our own experimental time series data and find high agreements (Supplementary Fig. 7 (32 genes vs. BG)). Interestingly, these identified metabolite-gene regulatory pairs are largely consistent with the global growth rate-regulated promoter activation from Kochanowsk et al. (Supplementary Data 6)[29,51].

The identified 32 genes are primarily located in the PPP and pyruvate metabolism in KEGG pathways (Fig. 6a), which are associated with cell growth for biosynthesis: generating NADPH and pentoses toward nucleotide and amino acid biosynthesis[52], instead of being in energy-producing pathways (TCA and glycolysis), which agrees with

the target pathway (reactions) of global cell state regulator: cAMP-Crp[33]. These results suggest that the expression of the genes in growth-associated pathways could be represented as a linear combination of the concentrations of biomass composition.

We then construct a transcriptional regulation-enabled linear kinetic model, i.e., Decrem integrated with global regulation, using the identified global growth state-regulated metabolic reactions (genes) based only on the concentration of corresponding biomass composition (BG) and metabolites (see Methods; Supplementary Note 3). To that end, we concentrate on the reactions catalyzed by the 32 identified metabolic genes. These genes are coordinated and vary with their

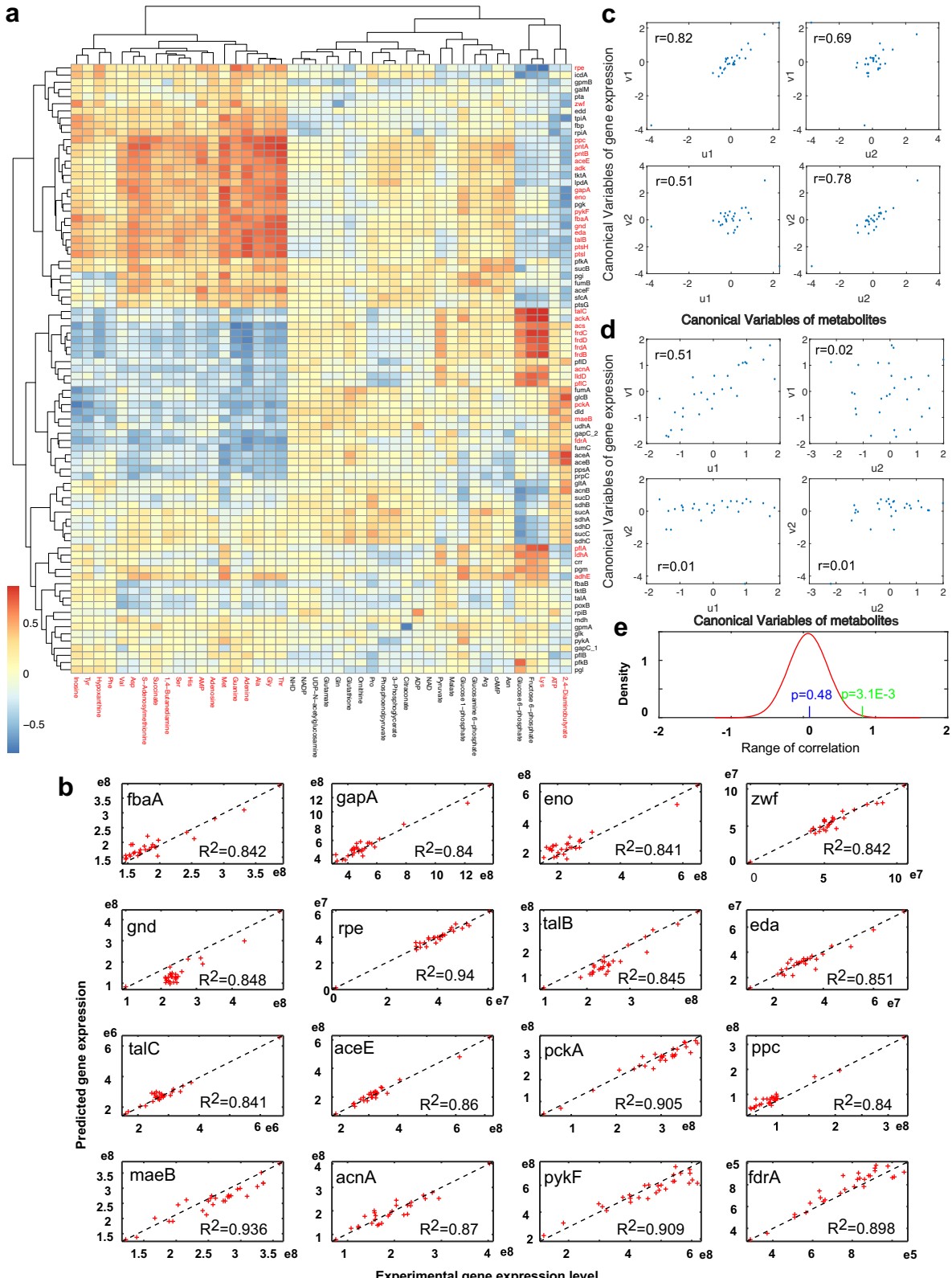

**Fig. 5 | The analysis of global regulatory metabolites-based gene transcription of E. coli. a** The correlations of regulator metabolites and gene expression in central metabolism; the column and row names in red are BG metabolites and identified globally regulated genes, respectively. **b** The predicted expression of 16 genes using the 23 biomass-associated metabolites by the partial least squares regression. **c, d** Canonical correlation analysis between BG (**c**) and PG (**d**) metabolites with the PLSR-identified genes. **e** The distribution of correlations between the measured and predicted gene expression levels using the metabolites from 10,000 random samplings (red curve), with the statistical significance levels according to the bio-mass metabolites group (green line) or the precursor metabolites group (blue line). One-sided *t*-test was used (random correlation samples *n* = 10,000, *p* = 3.1E−3 for BG and *p* = 0.48 for PG). Source data are provided as a Source Data file.

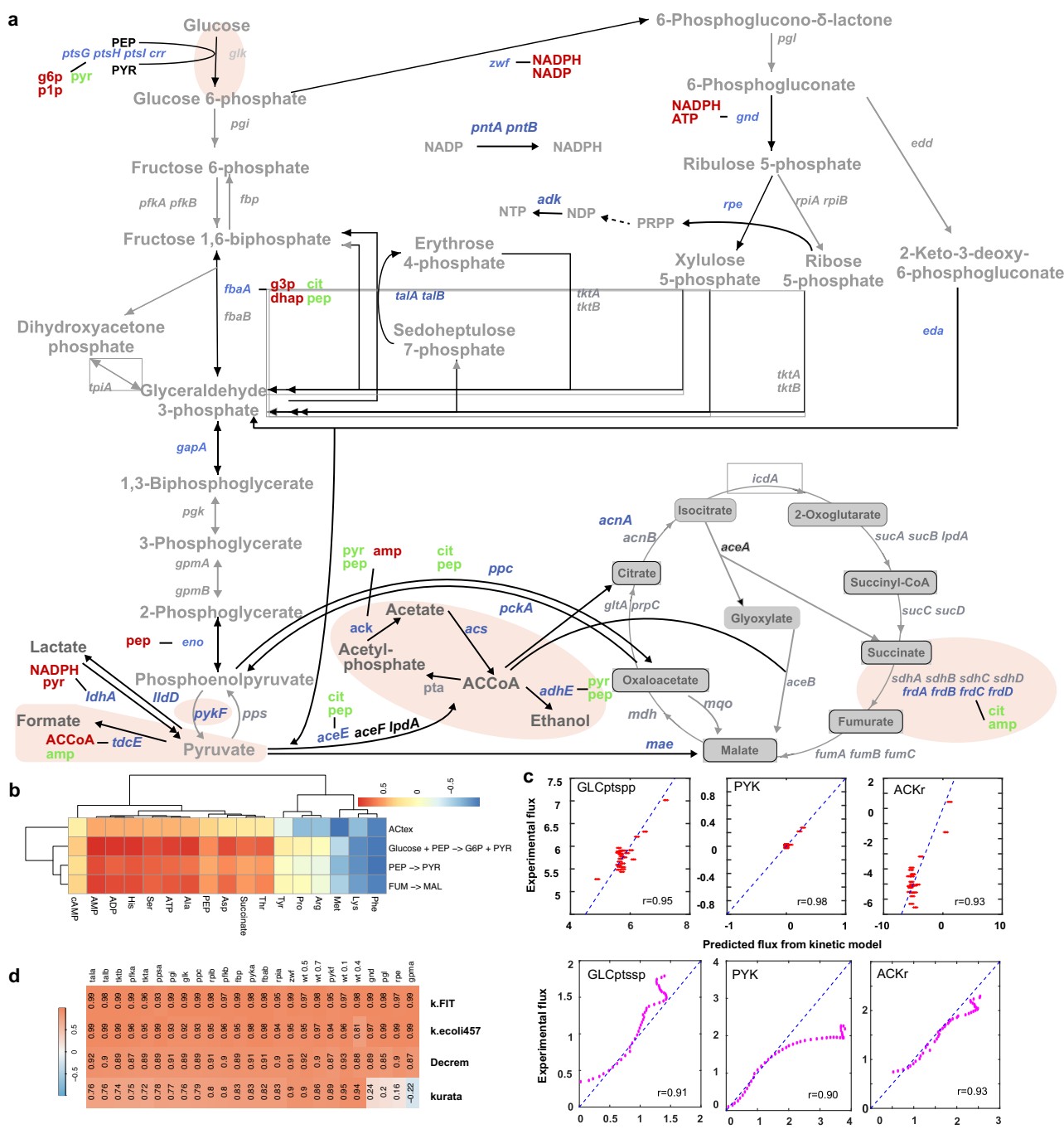

**Fig. 6 | The analysis of the metabolite-based kinetic regulation model of *E. coli*.**
**a** A map showing regulator metabolites and the genes regulated by the biomass-constituent metabolites in the central metabolism. The meanings of different backgrounds and colors: red and green metabolites are each gene inhibitors and activators; the identified 32 global growth-associated genes are in blue. The brown background indicates the identified key regulated reactions used to build the kinetic Decrem. **b** The correlation heatmap between the BP metabolites and identified global-regulated reactions. **c** Predicted kinetic fluxes on the training set (red) and the test set (purple) for three global regulatory reactions according to metabolite-determined linear Michaelis-Menten kinetics. ACKr acetate kinase, GLCptspp D-glucose transport via PEP:Pyr PTS (periplasm), PYK pyruvate kinase. **d** The correlation heatmap between the kinetic predicted flux and ¹³C FMA fluxes for identified global-regulated reactions, by the k-FIT[53], k-ecoli457[34], Decrem kinetic reactions, and kurata[35]. Source data are provided as a Source Data file.

fluxes (Supplementary Fig. 9), which are supported by well-established global regulatory mechanisms, e.g., the targets of cAMP-Crp activated phosphotransferase system (PTS)[31,33] which directly coordinate amino acid and carbohydrate uptake[31]. Five reactions, Hexokinase, Acetate kinase, Pyruvate dehydrogenase, Fumarate reductase, and Alcohol dehydrogenase, are identified as global PTS regulated. We then validate whether the five reactions are correlated with each specific growth rate using ¹³C-fluxes of a multi-strain dataset (including

69 strains varying from metabolic gene mutant to time series from different studies (Supplementary Data 7) and find a significantly high correlation of them (Fig. 6b). A specific advantage of the identified globally regulated reactions is the ability to get rid of local regulation influences with some dynamic enzyme activators or inhibitors. This reduces the regulatory complexity of reaction kinetics by ignoring the non-linear term of the Michaelis-Menten equation, as fluxes are mainly dominated by the global growth state (this property is easy to expand

to other non-model organisms) (see Methods). With the biomass composition-based linear representation of globally regulated genes, we can reformulate globally regulated enzyme kinetics as a form only related to metabolite concentration, which relieves the necessity of paired multi-omics data for canonical kinetic modeling.

Next, we model and evaluate the reformulated linear kinetics by optimizing the identified growth-associated key regulated fluxes in *E. coli* central metabolism. The predicted kinetic fluxes for global regulated reactions display high consistency with experimental measurements through [13]C isotope tracing (Fig. 6c), with *r* values being 0.95, 0.97, and 0.93 (*r* values are 0.91, 0.93, and 0.9 for the test dataset) for the reactions of glucose transport, pyruvate kinase, and acetate kinase, respectively (Fig. 6c, Supplementary Fig. 10, and Supplementary Data 7). These fluxes are also utilized to assign key kinetic fluxes in genome-scale metabolic flux prediction below (see Methods). Strikingly, our linear approximate kinetics only involves several global cell state-regulated reactions to constrain the flux distribution of Decrem, which still achieves good performance for flux prediction with the complex and large-scale kinetic models (Fig. 6d)[34,35,53].

### Growth rate estimation for *E. coli* genome-scale gene deletion strains using Decrem integrated with global regulation kinetics

We apply Decrem constructed above to predict the growth rates of *E. coli* genome-scale single-gene deletion mutants, using a dataset in which the growth rates and the concentrations of over 7000 metabolites have been experimentally measured[54]. A total of 1030 mutants with genes involved in metabolism are selected for growth analysis (Supplementary Note 4). We first examine the growth rates predicted by the methods incapable of global regulation, pFBA, MOMA, and Decrem without external flux constraints. As expected, poor results are produced, with low correlations with experimentally measured growth rates and *r* values of 0.127, 0.103, and 0.281 for pFBA, MOMA, and Decrem, respectively (Supplementary Fig. 11).

Next, we construct the global regulated linear kinetic fluxes of five identified reactions in central metabolism with predictions (based on metabolite concentrations) for each of 1030 mutants to approximate the mutant gene-specific metabolic state, e.g., the branch points of glycolysis and PPP, the flux allocation downstream of pyruvate metabolism, and the growth-associated secretion (see Methods; Fig. 6a and Supplementary Data 8). Using the GSMs integrated with kinetic fluxes, the growth rates of the 1030 mutants are estimated with six methods: Decrem, pDecrem, FBA, pFBA, RELATCH, and REPPS. The results show that all six kinetic methods have significantly improved predictions compared to the kinetic-free methods (Fig. 7a). Among them, Decrem and pDecrem produce the highest correlations with the empirical growth rates (*r* = 0.731 and 0.743 vs. 0.421, 0.685, 0.474, and 0.509) (Supplementary Data 8).

We then demonstrate the explanatory power of Decrem in interpreting the observed growth rates of mutants with corresponding (altered) flux distributions. For that, we calculate the distribution of pathway-specific fluxes across mutant strains, defined as the accumulated flux (AF) [13]C of each pathway (the accumulated sum of all nonzero fluxes in a pathway for each mutant strain) (Supplementary Note 4). The correlations between the AFs and growth rate for all strains show that Decrem quantifies the largest number of growth-related pathways that we curate from the literature compared to the other methods. For instance, many well-known pathways for cell growth—glutamate, nucleic acid, and most amino acid metabolic pathways—are 'detected' only by Decrem with significant correlation coefficients (Fig. 7b). Interestingly, although the globally regulated kinetic pFBA (also pDecrem) method can predict the growth rates with relatively high accuracy, the corresponding AFs cover only a few of the curated growth-associated pathways. Moreover, the analyses of the pathway-specific accumulated growth rates (AG; the accumulated sum of growth rates of strains in which the mutated genes are located in the

same pathway) suggest that Decrem-predicted distributions of AGs through all metabolic pathways are highly consistent with the experimentally measured AG distributions (Fig. 7c). FBA reaches similar levels of accuracy, but the predictions by pFBA only cover AGs, which are weakly influenced by gene knockouts and shrink the AGs to zero for the pathways containing knockout genes with strong growth effects (Fig. 7c). Such biases are an intrinsic property of the L1 norm-based pFBA (and pDecrem) method[54,55], despite the relatively good fit of correlations.

We further examine the flux variance distribution in each mutant for their changed growth rates. PCA of the fluxes predicted by Decrem across the 1030 mutants is shown in Fig. 7d (Supplementary Note 4). The distributions of the top two PCs indicate that the primary flux variances come from the decoupled LBRs, compared to the uncoupled reactions: 1.461 vs. 1.10 on average for PC1 (*t*-test, *p* = 3.04E−20); 1.56 vs. 1.07 for PC2 (*t*-test, *p* = 2.86E−25). This is consistent with the high robustness of the central metabolism[16,56] (primarily consisting of LBRs). Furthermore, we suspect that the growth effects of deleted genes encoding enzymes for the reactions within the SLBs would be more pronounced than the effect of genes encoding enzymes for uncoupled reactions. Indeed, this is confirmed by the analysis of the reaction type-based growth rates—the complex LBRs (the number of element reactions of associated linear basis vector > 1), the simple LBR (the number of element reactions of associated linear basis vector = 1) and the uncoupled reactions (Supplementary Note 4)—and the average growth rates are 0.666 (most impacted), 0.773 and 0.813 (least impacted) h[-1] (one-way ANOVA; *p* = 7.71E−28) (Fig. 7e). Finally, we examine the cause for the observed differences among the flux variances, the number of simple cycles, and the enzyme properties of LBRs. The flux variances are primarily explained by the multimeric enzymes and the topologically highly connected LBRs: LBRs are involved in a large number of simple cycles and few element reactions (Fisher's exact test, *p* = 2.34E−21 and 0.0051, respectively) (Fig. 7f). Therefore, the topological vulnerability of these reactions will result in functional variability.

## Discussion

We reconstructed a GSM model, Decrem, by identifying and incorporating local topologically decoupled reactions using SLB decomposition and by incorporating metabolic global regulation by metabolites into GSMs, which approximates the kinetic fluxes of cell state-regulated key reactions to constrain the feasible region of optimal flux distribution. Decrem effectively reduces the requirements for multi-omics data for genome-scale metabolic kinetic models. Compared to existing methods, Decrem demonstrates superior performance in predicting metabolic fluxes in three model organisms and growth rates in genome-scale knockout strains of *E. coli*. Therefore, it is an effective model for accurately depicting metabolic responses and exploring the self-adapting regulation mechanism of cellular perturbation.

By applying SLB decomposition, the (coupled) element reactions within identified SLBs display high coexpression among multiple growth conditions, indicating coordinated activation of topologically highly coupled reactions. Interestingly, similar approaches have been applied in identifying the non-redundant local functional units of metabolism, i.e., the minimal metabolic pathway or flux tope[38,57]. A topological orthogonality principle has been successfully used to design bioengineering strains with minimal interaction between desired product-associated pathways and metabolic components related to biomass synthesis[58]. In addition, several specific topological constraint treatments, such as removing the thermodynamically infeasible loops and decoupling two desired phenotypes, have been applied to GSMs to improve their metabolic production in recent studies[59,60]. But Decrem is the first genome-scale topologically decoupled metabolic model for general applications, which clearly demonstrates how the topological preference of a metabolic network can guide the metabolic flux distribution.

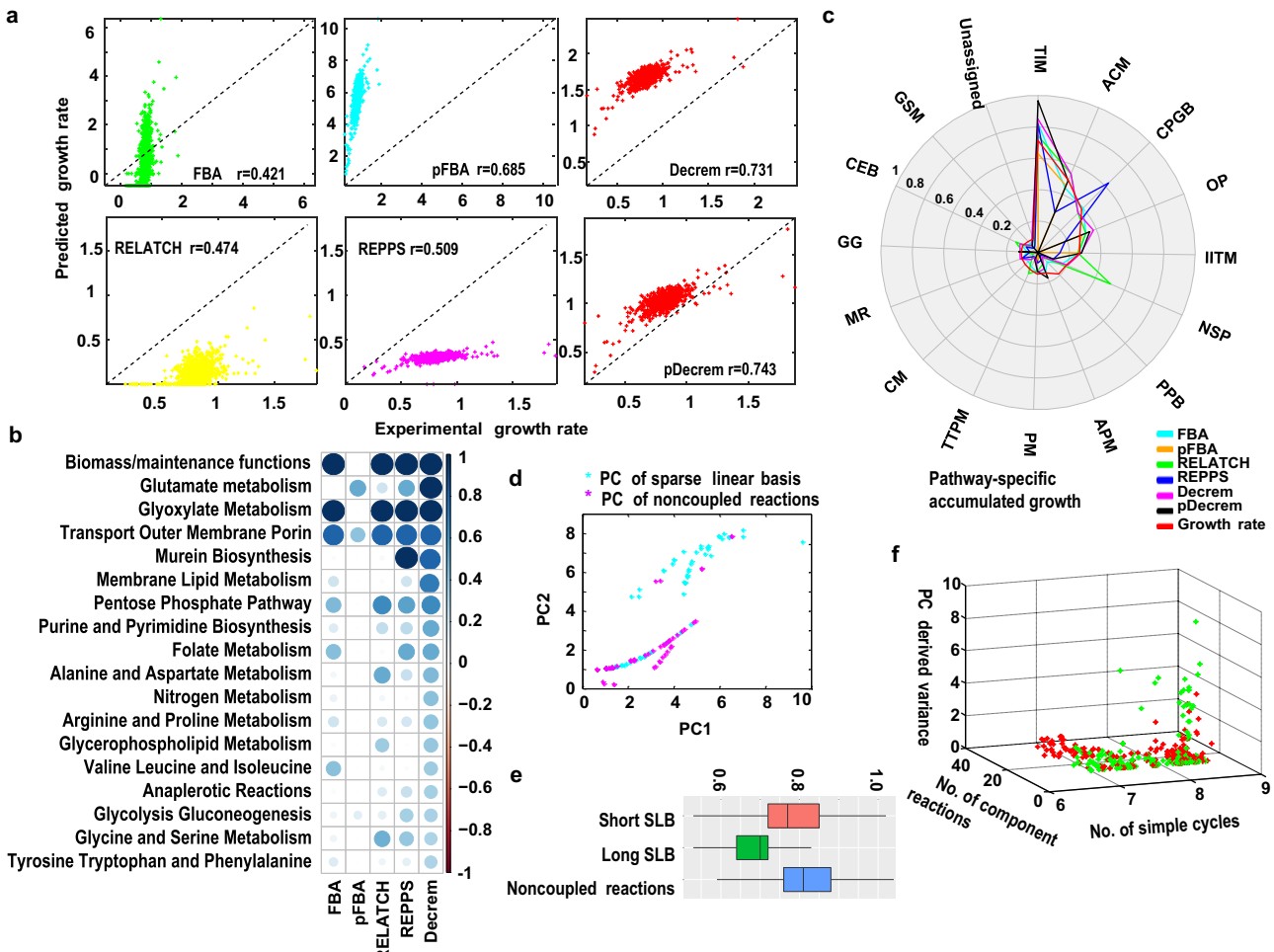

**Fig. 7 | The growth rate analysis on the genome-scale gene deletion strains.** **a** The comparison of measured and predicted growth rates by the pDecrem, Decrem, FBA, pFBA, RELATCH, and REPPS on 1030 *E. coli* mutants. **b** The correlation between the measured growth rates and the predicted accumulated fluxes through the identified growth-associated pathways. **c** The comparison of the predicted accumulated growth rates through the pathways where the mutants belong, according to the six test methods. **d** The top two PCs revealed differences in flux variances between the coupled reactions (cyan dots) ($n = 789$) and the non-coupled reactions (purple dots) ($n = 1458$) predicted by Decrem according to the 1030 strains. (unpaired two-tailed *t* test, $p = 3.04E-20$ for PC1, $p = 2.86E-25$ for PC2). **e** The comparison of growth rate distribution among topological components where the mutants belong: the complex LBR (the number of element

reactions of corresponding sparse linear basis vector > 1) ($n = 208$), the simple LBR (the number of element reactions of corresponding sparse linear basis vector = 1) ($n = 268$) and the non-coupled reactions ($n = 360$) (one-way ANOVA test, $p = 7.71E-28$). The black center line denotes the median value (50th percentile), while the green and orange boxes contain the 25th to 75th percentiles of the dataset. The black whiskers mark the 5th and 95th percentiles, and values beyond these upper and lower bounds are considered outliers, marked with black bars. **f** The flux variance of LBR based on the multimeric classification: the multi-subunit enzymes (green dots) ($n = 218$) and single-unit enzymes (red dots) ($n = 258$), the number of simple cycles and the length (the number of element reactions) of LBR (Fisher's Exact Test, *p* values = 2.34E−21 and 0.0051). Source data are provided as a Source Data file.

To explore effective metabolic dynamic robustness or adaption to internal and external perturbations, metabolic kinetic models have attracted great attention by combining fluxomics, metabolomics, and transcriptomics into a unified framework[34,61,62]. However, the construction of the GSM kinetic model is obstructed by the limited knowledge about kinetic parameters, e.g., $K_m$, $K_{cat}$, and the scarcity of metabolic regulators and paired genome-scale multi-omics data[34,63]. To date, the largest metabolic kinetic model, k-ecoli457 of *E. coli*, contains only 457 reactions, 337 metabolites, and 295 substrate-level regulatory interactions according to the computationally predicted kinetic parameters[34]. Alternatively, by integrating the metabolite-TF regulatory regime, Decrem integrated with global regulation kinetics can predict growth rates and the corresponding fluxes. An advantage of our kinetic Decrem is that it only needs experimental concentrations of identified essential metabolites, which serve as the key indicators of metabolic states and directly regulate enzyme activities or gene expression. Although some metabolites (e.g., those in

glycolysis and the TCA cycle) have long been known to regulate enzyme activities as detailed in biochemistry textbooks, not many are known on the whole genome scale, even in model organisms. On the one hand, the regulatory metabolites of specific pathways in central metabolism prefer to frequently interact with the catalytic enzymes by activating or inactivating the functional domains to synchronously adjust the fluxes[63,64]. On the other hand, several studies have revealed that ~70% of the total variance in the promoter activity of central metabolic genes of *E. coli* can be explained by growth rate-derived global transcriptional regulation across multiple mutant strains[29,65]. These findings suggest a potential relationship between the concentration of biomass-constituent metabolites and the expression of metabolic genes in the regulons of TFs. This relationship is verified by a recent study in which the identified metabolite concentrations are predicted by quantitative proteomics data[45]. Overall, both the topology of metabolic networks and regulatory metabolites are utilized to identify coactivated or key

regulated reactions, which produces a minimal set of regulatory constraints to develop the genome-scale kinetic models.

Compared to other methods tested on three model microorganisms, Decrem is not only excellently performing in recovering the real flux distributions with high accuracy in a wide range of strains but also presents excellent predictions and explanations of the observed growth rate. This demonstrates the strong capability of Decrem to approximate the real intracellular state and to be used for designing high-yield mutant strains in bioengineering and synthetic biology. Decrem shows that there is a strong influence of metabolic network topology in the prediction of flux distributions; this phenomenon is found both in the reconstructed SBRs of Decrem and in the two versions of *Yeast* metabolic models. A possible explanation for this observation is that the optimal flux distribution of a metabolic network is strongly determined by its topology, and the rewired or perturbed network structure will have different feasible regions. Moreover, Decrem could be applied to elucidate important regulatory branch points and the self-adaption of regulation mechanisms for the knockout strains, which is helpful for accurately predicting the potential target genes/reactions for designing bioengineering strains.

The main limitation of Decrem is that it requires a medium-scale set matched [13]C-MFA flux paired with metabolite concentration data to construct reasonable kinetic models. However, the well-constructed kinetic models are convenient for transfer to any other applications. Overall, Decrem, a local topology and global regulatory network-reinforced metabolic analysis model, can accurately predict phenotypes and uncover the complex regulation of cell metabolism.

## Methods

### Topology-decoupled reconstruction of the metabolic model

The original metabolic network consists of coupled coactivated reaction cycles (e.g., TCA cycle) and simple linear chain reactions (e.g., biosynthetic reaction chains). We developed a GSM model, Decrem, to capture the contribution of the coactivated and coupled reactions while preserving the consistency of the linear components. The detailed framework for our model is described below (Supplementary Fig. 1).

Step 1: Identifying reaction cycle-based coupled substructures. We first defined the bipartite graph representation of the metabolic network as $\mathbf{G}(\mathbf{V}_{m+n}, \mathbf{E})$, where the node set $\mathbf{V}_{m+n}$ includes both the $m$ metabolite nodes and $n$ reaction nodes, and the edge set $\mathbf{E}$ includes all the interactions between the metabolite and reaction nodes. And built the similarity matrix $\mathbf{A}_{n \times n}$ for $n$ metabolic reactions based on the number of topological simple cycles of $\mathbf{G}$. Specifically, the element $a_{ij}$ of $\mathbf{A}_{n \times n}$, which indicates the similarity between reactions $i$ and $j$ in the metabolic network, is defined as the number of simple directed cycles passing through the paired reaction nodes $(v_i, v_j)$ in $\mathbf{G}$. According to the similarity matrix $\mathbf{A}_{n \times n}$ built above, the BestWCut clustering algorithm[36] was used to identify the dense substructures (also known as network communities) consisting of highly coupled reaction cycles. Here, the substructures are denoted as $\mathbf{C} = \{\mathbf{C}_k | k = 1 \ldots K\}$, where $\mathbf{C}_k$ is a subset of reaction index set $[n] = 1, \ldots, n$, $k$ is the index of substructures $\mathbf{C}_k$, and $K$ is the total number of substructures. If we define $D_i = \sum_{j \in [n]} a_{ij}$ as the weighted out-degree of reaction node $v_i$, $i \in [n]$, then the cluster degree $D_k$ for subnetwork $\mathbf{C}_k$ could be defined as follows:

$$D_k = \sum_{i \in \mathbf{C}_k} D_i \tag{1}$$

The generalized weighted cut (*WCut*) associated with $\mathbf{C}$ is obtained by minimizing $WCut(\mathbf{C})$:

$$WCut(\mathbf{C}) = \sum_{k=1}^{K} \sum_{k' \neq k} \frac{Cut(\mathbf{C}_k, \mathbf{C}_{k'})}{D_k} \tag{2}$$

where

$$Cut(\mathbf{C}_k, \mathbf{C}_{k'}) = \sum_{i \in k} \sum_{j \in k'} a_{i,j} \tag{3}$$

and $\mathbf{C}_{k'}$ is the complement set of the cluster $\mathbf{C}_k$.

Step 2: Reconstructing the decoupled representation of the identified substructure using the sparse linear basis. Inspired by the minimal metabolic pathways[38], we represented the highly coupled substructures with the SLBs of the null space of the corresponding stoichiometric matrix, which is biologically explained as the minimal and indecomposable coupled components, and satisfies the constraint of thermodynamics and mass balance of element reactions. Unlike infinite ordinary linear basis vectors of the null space of the stoichiometric matrix, there is a unique and globally optimal sparsest basis group of the null space[38,66]. Briefly, the orthonormal null space $\mathbf{N}_{\mathbf{C}_k}$ is initially defined by singular value decomposition (SVD) for the stoichiometric matrix $\mathbf{S}_{\mathbf{C}_k}$ of the subnetwork $\mathbf{C}_k$. Here, additional artificial exchange reactions are introduced in $\mathbf{S}_{\mathbf{C}_k}$ to maintain the mass balance of reactions in the subnetwork $\mathbf{C}_k$ (we explain that those artificial exchange reactions only are used to assist the SLB decomposition, and will be removed in the next step; more details can be found in Supplementary Note 5). Then, the column vectors of the orthonormal null space $\mathbf{N}_{\mathbf{C}_k}$ are iteratively replaced by the minimal element reactions that span the removed subspace of vectors. This process is repeated until all the nonzero entries in $\mathbf{N}_{\mathbf{C}_k}$ are converged on a minimum[38]. Here, we utilized the advantage of sparse regularization of the null space $\mathbf{N}_{\mathbf{C}_k}$ to solve the minimum L1-norm of the null space $\mathbf{N}_{\mathbf{C}_k}$ of $\mathbf{S}_{\mathbf{C}_k}$[66]. The detailed process is showcased in Supplementary Note 5, in which $\mathbf{N}_{\mathbf{C}_k}^{\mathbf{S}}$ is a minimal sparse basis representation of $\mathbf{N}_{\mathbf{C}_k}$ in at most $2r_k$ linear programming optimization runs (where $r_k = l_k - rank(\mathbf{S}_{\mathbf{C}_k})$, $l_k$ is the number of columns (reactions) in $\mathbf{S}_{\mathbf{C}_k}$). If we assume $\mathbf{x} \in \mathbb{R}^n$ and then each linear programming problem can be formulated as follows:

$$\min_{\left(\mathbf{x}, \mathbf{v}_{\mathbf{C}_k}^m\right)} \sum_i \mathbf{x}_i \tag{4}$$

$$\text{s.t.} \; \mathbf{S}_{\mathbf{C}_k} . \mathbf{v}_{\mathbf{C}_k}^m = 0 \tag{5}$$

$$\mathbf{v}_{\mathbf{C}_k}^m \leq \mathbf{x} \tag{6}$$

$$-\mathbf{v}_{\mathbf{C}_k}^m \leq \mathbf{x} \tag{7}$$

$$\mathbf{lb} \leq \mathbf{v}_{\mathbf{C}_k}^m \leq \mathbf{ub} \tag{8}$$

$$\mathbf{w}^T . \mathbf{P}_{\mathbf{C}_k}^{N_m} . \mathbf{v}_{\mathbf{C}_k}^m > \zeta \; \bigvee \; \mathbf{w}^T . \mathbf{P}_{\mathbf{C}_k}^{N_m} . \mathbf{v}_{\mathbf{C}_k}^m < -\zeta \tag{9}$$

where $\mathbf{v}_{\mathbf{C}_k}^m$ is the SLB of the null space of $\mathbf{S}_{\mathbf{C}_k}$ at the $m^{\text{th}}$ run, and $\mathbf{lb}$ and $\mathbf{ub}$ are the lower and upper bound of $\mathbf{v}_{\mathbf{C}_k}^m$, respectively. The constraint of formula (9) ensures that $\mathbf{v}_{\mathbf{C}_k}^m$ is linearly independent of the previous $m-1$ SLBs, and $\mathbf{w}$ represents a vector of random weights. Here, we employed uniform random weights, and $\zeta$ is a small positive constant, e.g., $1.0^{-3}$. $\mathbf{P}_{\mathbf{C}_k}^{N_m}$ is a projection matrix onto the sparse null space $\mathbf{N}_{\mathbf{C}_k}^{\mathbf{S}_m}$, and $\mathbf{N}_{\mathbf{C}_k}^{\mathbf{S}_m} = [\mathbf{v}_{\mathbf{C}_k}^1, \mathbf{v}_{\mathbf{C}_k}^2, \ldots, \mathbf{v}_{\mathbf{C}_k}^{m-1}]$. More details of this process are provided in Supplementary Note 5.

With the $\mathbf{N}_{\mathbf{C}_k}^{\mathbf{S}} = [\mathbf{v}_{\mathbf{C}_k}^1, \mathbf{v}_{\mathbf{C}_k}^2, \ldots, \mathbf{v}_{\mathbf{C}_k}^{r_k}]$ as the assembled representation of SLBs of $\mathbf{S}_{\mathbf{C}_k}$, assembled as:

$$\mathbf{S}_{\mathbf{C}_k}^{IBR} = \mathbf{S}_{\mathbf{C}_k}^* . \mathbf{N}_{\mathbf{C}_k}^{\mathbf{S}^*} \tag{10}$$

where $\mathbf{S}^*_{\mathbf{C_k}}$ and $\mathbf{N}^{S^*}_{C_k}$ are derived from $\mathbf{S}_{\mathbf{C_k}}$ and $\mathbf{N}^S_{C_k}$ after removing the artificial exchange reactions, respectively, and $IBR$ indicates the reconstructed independent LBR.

Step 3: Establishing prediction for the decoupled Decrem metabolic model.

We reformulated FBA to adapt to the reconstructed decoupled metabolic network $\mathbf{S}^{IR}$ by Decrem. The key objective is to determine the flux bounds for each LBR:

$$\max_{\mathbf{v}^{IR}} \mathbf{c}\mathbf{v}^{IR} \tag{11}$$

$$\text{s.t.} \quad \mathbf{S}^{IR}.\mathbf{v}^{IR} = 0 \tag{12}$$

$$\mathbf{S}^{IR} = \left[\mathbf{S}^{NC}, \mathbf{S}^{IBR}_{C_1}, \ldots, \mathbf{S}^{IBR}_{C_K}\right] = \left[\mathbf{S}^{NC}, \mathbf{S}^*_{C_1}\mathbf{N}^{S^*}_{C_1}, \ldots, \mathbf{S}^*_{C_K}\mathbf{N}^{S^*}_{C_K}\right] \tag{13}$$

$$\mathbf{v}^{IR} = \left[\mathbf{v}^{NC}, \mathbf{v}^{IBR}_{C_1}, \ldots, \mathbf{v}^{IBR}_{C_K}\right]^T \tag{14}$$

$$\mathbf{v}^{IBR}_{\mathbf{C_k}} = \left[\mathbf{v}^1_{\mathbf{C_k}}, \ldots, \mathbf{v}^{r_k}_{\mathbf{C_k}}\right]^T \tag{15}$$

$$\mathbf{lb}^{NC} \leq \mathbf{v}^{NC} \leq \mathbf{ub}^{NC} \tag{16}$$

$$\max\left(f\left(lb^i_{\mathbf{C_k}}, NZ\left(\mathbf{N}^i_{\mathbf{C_k}}\right)\right)./f_N\right) \leq \mathbf{v}^i_{C_k} \leq \min\left(f\left(ub^i_{\mathbf{C_k}}, NZ\left(\mathbf{N}^i_{\mathbf{C_k}}\right)\right)./f_N\right) \tag{17}$$

$$f_N = f\left(\mathbf{N}^i_{\mathbf{C_k}}, NZ\left(\mathbf{N}^i_{\mathbf{C_k}}\right)\right) \tag{18}$$

where $\mathbf{c}$ and $\mathbf{v}^{IR}$ represent the objective function and optimal metabolic flux of $\mathbf{S}^{IR}$, respectively. The superscript $IR$ represents the linearly independent reaction-derived metabolic network, and $NC$ represents the noncoupled reactions (which are composed of linear reaction chains) of the original metabolic network. $r_k$ is the number of columns of $\mathbf{N}^{S^*}_{\mathbf{C_k}}$, and $K$ is the total number of highly coupled reaction subnetworks identified in *step* 1. $\mathbf{N}^i_{\mathbf{C_k}}$ is the $i$ th column of $\mathbf{N}^{S^*}_{\mathbf{C_k}}$, and $lb^i_{\mathbf{C_k}}$ and $ub^i_{\mathbf{C_k}}$ are the lower and upper bounds of the reaction indicated by $\mathbf{N}^i_{\mathbf{C_k}}$, respectively. $\mathbf{v}^{IBR}_{\mathbf{C_k}}$ is the flux vector of all the LBR of subnetwork $\mathbf{C}_k$, and $\mathbf{v}^i_{\mathbf{C_k}}$ is the $i$ th flux of $\mathbf{v}^{IBR}_{\mathbf{C_k}}$. Among them, $i$ ranges from 1 to $r_k$, and $k$ ranges from 1 to $K$. Then, the fluxes of reactions in the original metabolic network (element reactions) of linear basis vectors will be recovered by the formula $\mathbf{N}^{S^*}_{\mathbf{C_k}}.\mathbf{v}^{IBR}_{C_1}$ according to the optimal solution of Decrem outlined above.

The function $NZ(.)$ takes the index of nonzero elements of the input vector, and the function $f(\mathbf{v}, I)$ takes elements indexed by the input indicator $I$ from the input vector $\mathbf{v}$. Therefore, $f(\mathbf{N}^i_{\mathbf{C_k}}, NZ((\mathbf{N}^i_{\mathbf{C_k}}))$. represents the nonzero partition coefficient of each element reaction composed of the $i$th SLB of subnetwork $\mathbf{C}_k$, which is indicated by the nonzero terms of $\mathbf{N}^i_{\mathbf{C_k}}$. The formula $f(lb^i_{\mathbf{C_k}}, NZ(\mathbf{N}^i_{\mathbf{C_k}}))$ represents the lower bounds of the element reactions composed of the $i$th SLB of the subnetwork $\mathbf{C}_k$, and $f(ub^i_{\mathbf{C_k}}, NZ(\mathbf{N}^i_{\mathbf{C_k}}))$ represents the upper bounds. In summary, Decrem forces the metabolic fluxes of highly coupled reactions to be incorporated into optimization by representing them as independent linear basis vectors. On the basis of Decrem, we proposed parsimonious Decrem (pDecrem) with parsimonious FBA optimizer. More details of the model are provided in Supplementary Note 6.

## Transcription regulation mechanism
We developed a mechanistic basis model to link the kinetics of transcription to metabolite regulators based on the gene regulatory model[51]. From this model, we can get a linear regulation model between the central metabolism gene activity and local regulators, as well as the global regulators.

$$\log\left(E_g\right) \approx \alpha_g \log(R) + \sum_{i=1}^{K} \beta_{gi} \log(M_{gi}) \tag{19}$$

where $E_g$ represents the expression $E$ of gene $g$, $R$ indicates the given growth rate, $M_{gi}$ represents the $i$th of $K$ metabolite regulators, and $\alpha_g$, $\beta_{gi}$ represent the corresponding coefficients. According to the biomass reactions, we can represent the growth rate as follows:

$$R = \lambda \prod_{j=1}^{N}(1 + M_{bj}/K_{mj})^{\theta_j} \tag{20}$$

where the $M_{bj}$ represents the $j$ th of $N$ biomass metabolites, $K_{mj}$ is a cell state-related kinetic parameter, $\lambda$ and $\theta_j$ are the reaction coefficients. Following a previous study[51], we approximate $\log(1 + M_{bj}/K_{mj})$ with $\log(M_{bj}/K_{mj})$, then we take the logarithm of the above equation and approximate it as follows:

$$\begin{aligned}
\log(R) &= \sum_{j=1}^{N} \theta_j \log\left(1 + M_{bj}/K_{mj}\right) + \log(\lambda) \\
&\approx \sum_{j=1}^{N} \theta_j \log\left(\frac{M_{bj}}{K_{mj}}\right) + \log(\lambda) \\
&= \sum_{j=1}^{N} \theta_j \log\left(M_{bj}\right) + c
\end{aligned} \tag{21}$$

So,

$$\log\left(E_g\right) \approx \alpha_g \sum_{j=1}^{N} \theta_j \log\left(M_{bj}\right) + \sum_{i=1}^{K} \beta_{gi} \log\left(M_{gi}\right) + b \tag{22}$$

where $M_{bj}$ indicates the biomass metabolites and $M_{gi}$ represents the TF regulating metabolites, then we got an approximate quantitative relationship between gene expression and metabolite concentration in the central metabolism. Furthermore, we can identify the dominant regulators of transcription according to the multi-omics data analysis under multiple strains.

## Gene expression estimation
Depending on the above transcription regulation mechanism, 45 candidate global and local regulatory metabolites of *E. coli* are collected through KEGG pathway analysis and a literature survey[48–50]. These candidates are then categorized into two clusters by hierarchical clustering analysis over the gene expression profile and their metabolite concentrations across 24 mutant strains. Then, the possible regulatory relationship between two identified metabolite groups and the expression of 85 genes are inferred by PLSR[67], which selects the nonredundant and independent factors to maximize the correlation of response variables using stepwise principal component regression. Furthermore, PLS is used to discover the fundamental quantitative relations between two observation variable sets, and the general underlying model of multivariate PLS is described as follows:

$$\mathbf{X} = \mathbf{T}\mathbf{P}^\mathbf{T} + \mathbf{E} \tag{23}$$

$$\mathbf{Y} = \mathbf{U}\mathbf{Q}^\mathbf{T} + \mathbf{F} \tag{24}$$

where $\mathbf{X}$ is an $n \times m$ matrix of predictors (metabolite concentrations), and $\mathbf{Y}$ is an $n \times p$ matrix of responses (gene expression). $\mathbf{T}$ and $\mathbf{U}$ are $n \times l$ matrices and projections of $\mathbf{X}$ (the $\mathbf{X}$ score, component or factor

matrix) and projections of **Y** (the **Y** scores), respectively. **P** and **Q** are $m \times l$ and $p \times l$ orthogonal loading matrices, respectively. Matrices **E** and **F** are the error terms, assumed to be independent and identically distributed standard normal random variables. The decomposition of **X** and **Y** was performed to maximize the covariance between **T** and **U**.

Step 2: The optimal strategy of the linearized kinetic model. In this section, we sought the simplified Eq. (27) representation based only on the associated metabolite concentrations. Firstly, we took the negative logarithmic operation of both sides of the Eq. (27) and reorganized the right-hand terms:

$$-\log(v([S],[P],[A],[I],[E])) = \underbrace{-\log([E])}_{\text{Enzyme term}} \underbrace{-\log\left(1-[P]/[S]\,e^{-\Delta_r G'^{\circ}/RT}\right) + \sum_v \log(1+[I_v])}_{\text{Metabolite associated terms}}$$

$$\underbrace{-\log(k^+) - \sum_v k_I^v}_{\text{Kinetic constants}} + \underbrace{\sum_u \log\left(1+\frac{k_A^u}{[A_u]}\right) + \log\left(1+\frac{k_m^s}{[S]}\left(1+\frac{[P]}{k_m^p}\right)\right)}_{\text{Nonlinear terms}}$$

(28)

Finally, the significant metabolite profile and corresponding explicable gene-metabolite regulatory relationships are filtered by setting the proper correlation threshold. The statistical test is built based on random sampling (Supplementary Note 3). The identified gene regulation is validated by the canonical correlation analysis[68] on our experimental data.

**The metabolite concentration-derived linearized kinetic model**
Step 1: The metabolic kinetic model. In this section, we derived a complete reversible rate law for arbitrary reactant stoichiometries. When considering the constraint of thermodynamics and metabolite regulation[63,69], we can rewrite the Michaelis–Menten kinetics for a reversible reaction S <=> P as follows:

$$v([S],[P],[A],[I],[E]) = [E]\frac{k^+[S]/k_m^s - k^-[P]/k_m^p}{1+[S]/k_m^s+[P]/k_m^p}\prod_u\frac{[A_u]/k_A^u}{1+[A_u]/k_A^u}$$
$$/\prod_v\frac{1}{1+[I_v]/k_I^v}$$

(25)

where $[E]$ is the concentration of enzyme active sites, $[S]$ and $[P]$ are the concentrations of substrates and products, $k_m^s$ and $k_m^p$ are the affinities of the reactants for this enzyme, $k^+$ and $k^-$ are the maximal forward and reverse catalytic rate constants, $[A]$ and $[I]$ are the concentrations of activators and inhibitors, and $k_a^{A_u}$ and $k_i^{I_v}$ are their corresponding affinities. The positive and negative terms in the numerator are associated with the forward and backward rates, respectively.

We next applied the metabolic thermodynamics constraints given by the Haldane relationship to simplify the term for the backward rate:

$$k^{eq} = \frac{k^+ k_m^p}{k^- k_m^s} = \frac{[P_0]}{[S_0]} = e^{-\Delta_r G^{\circ}/RT}$$

(26)

where $\Delta_r G^{\circ}$ is the standard Gibbs energy of the reaction (and does not depend on the enzyme parameters). Using this equality with the above rate law, we can obtain the following:

$$v([S],[P],[A],[I],[E]) = ([E]k^+)\frac{[S]/k_m^s(1-[P]/[S]e^{-\Delta_r G'^{\circ}/RT})}{1+[S]/k_m^s+[P]/k_m^p}$$
$$\prod_u\frac{[A_u]/k_A^u}{1+[A_u]/k_A^u}/\prod_v\frac{1}{1+[I_v]/k_I^v}$$

(27)

This model can be solved by collecting the corresponding kinetic parameters, enzyme expression, and metabolite concentrations. However, those matched data are often unavailable in practice, and the metabolic regulators often need to be discovered. An alternative is to approximate the optimal parameters using to the machine learning method. Specifically, the global cell growth state-regulated enzyme expression can be represented as the linear combination of the concentrations of biomass composition and TF regulators, which can be marked as $\log(E_g) \approx \alpha_g \sum_{j=1}^N \theta_j \log(M_{bj}) + \sum_{i=1}^K \beta_{gi} \log(M_{gi}) + b$, where $M_{bj}$ indicates the biomass metabolites, $M_{gi}$ represents the TF regulating metabolites in the "Transcription regulation mechanism" section. Specifically, the global regulated gene expression can be simplified as $\log(E_g) \approx \alpha_g \sum_{j=1}^N \theta_j \log(M_{bj})$ through the section of "Gene expression estimation". In addition, we reexamined the nonlinear terms of equation (28) based on the knowledge that systemic experimental analysis revealed that $[S]$ was $\geq k_m^s$ for almost all of the metabolites in the central metabolism of three model organisms[15]. Hence, we have $\frac{k_m^s}{[S]} \leq 1$, then we can get a linear kinetic formulation after several steps of derivation (Supplementary Note 7):

$$\log(v([S],[P],[A],[I],[E])) \approx \alpha_g \sum_{j=1}^N \theta_j \log\left(M_{bj}\right) + \log\left(1-[P]/[S]e^{-\Delta_r G^{\circ}/RT}\right)$$
$$+ \frac{[P]/[S]}{k^{eq}} + \sum_u\left(\frac{k_A^u}{[A_u]}\right) + constant$$

(29)

This result can be expanded to multi-substrate/multi-product reactions. When we neglect the infinitesimal of higher order, the identified regulators and optimal kinetic parameters in models (29) can be solved with linear regression. Subsequently, the optimal model is expanded to any other application.

Step 3: The linearized kinetic optimization of Decrem. Finally, the parameterized kinetic model is utilized to describe the growth-associated key-regulated reactions in central metabolism. The genome-scale flux distribution is predicted by the kinetic regulated flux-constrained Decrem method, i.e.,

$$\max \mathbf{c}\mathbf{v}_{obj}$$

(30)

$$\text{s.t. } \mathbf{S}.\mathbf{v} = 0$$

(31)

$$\mathbf{lb}^{NC} \le \mathbf{v}^{NC} \le \mathbf{ub}^{NC} \tag{32}$$

$$\max\left(f\left(lb^i_{C_k}, NZ\left(\mathbf{N}^i_{C_k}\right)\right)./ff_N\right) \le \mathbf{v}^i_{C_k} \le \min\left(f\left(ub^i_{C_k}, NZ\left(\mathbf{N}^i_{C_k}\right)\right)./f_N\right) \tag{33}$$

$$f_N = f\left(\mathbf{N}^i_{C_k}, NZ\left(\mathbf{N}^i_{C_k}\right)\right) \tag{34}$$

$$v^{KF}_j - \delta \le v_j \le v^{KF}_j + \delta \tag{35}$$

where $v^{KF}_j$ is the $j$-th kinetic flux of the $m$ key regulation reactions, and $\delta$ is the tolerance of kinetic fluxes. The $i, k, NC, lb^i_{C_k}, NZ\left(\mathbf{N}^i_{C_k}\right)$ can be found in *step* 3 of the section "Topology-decoupled reconstruction of the metabolic model". Among them, $i$ ranges from 1 to $r_k$, $j$ ranges from 1 to m, and $k$ ranges from 1 to $K$.

### E. coli culturing

Strain and culturing. *E. coli* strain BW25113 was grown in MOPS minimal medium (Teknova Inc, California, USA) with glucose at 2 g/L with shaking at 120 rpm at 37 °C. Aliquots of cells were collected at four growth states (timepoints): the beginning of the lag phase, the transition from lag to log phase, the mid-log phase, and the early stationary phase. Aliquots of cells were collected at each timepoint/growth state for RNA-seq and metabolomic profiling. Three replicates per growth condition and time point. No statistic methods used to predetermine sample size, no sample size calculation was performed, the sample was choosing by the growth state of *E. coli*.

### Transcriptomic analysis of E. coli

For RNA-seq, total RNA was extracted using the Qiagen RNeasy Mini kit (Qiagen Inc, MD, USA) following the manufacturer's instructions and sequenced on the Illumina Hi-seq 2500 platform. Raw reads were quality controlled using FASTQC and trimmed using Trimmomatic 0.39[70] with a quality score of 26. The read counts for each gene were analyzed using RSEM[71]. These raw data are deposited onto the NCBI Short Read Archive (SRA) database with Project accession PRJNA910919.

### Metabolomic analysis of E. coli

Frozen cells were broken on dry ice with the bead beater and kept in liquid nitrogen between homogenization and extraction[72]. Specifically, the extraction solvent was eisopropanol/acetonitrile/water at the volume ratio 3:3:2 and cooled to −20 °C prior to extraction. 1 ml of cold solvent per 20 mg of cells was added, vortexed for 10 s, and shaken at 4 °C for 5 min to extract metabolites and simultaneously precipitate proteins. Extracts were centrifuged for 20 min at −4 °C at 17,000 × g to remove the cell debris. Centrifuged extracts were analyzed by LC-MS/MS, with an Agilent 6495 triple quadrupole mass spectrometer (Agilent Technologies). Data were acquired using the following chromatographic parameters. Column: Restek corporation Rtx-5Sil MS (30 m length × 0.25 mm internal diameter with 0.25 μm film made of 95% imethyl/5%diphenylpolysiloxane). Mobile phase: Helium; Column temperature: 50–330 °C. Flow-rate: 1 mL min-1; Injection volume: 0.5 μL. Injection: 25 splitless times into a multi-baffled glass liner; Injection temperature: 50 °C ramped to 250 °C by 12 °C s⁻¹; Oven temperature program: 50 °C for 1 min, then ramped at 20 °C min-1 to 330 °C, held constant for 5 min.

Raw data files are preprocessed directly after data acquisition and stored as ChromaTOF-specific.peg files, as generic.txt result files and additionally as generic ANDI MS.cdf files. ChromaTOF vs. 2.32 is used for data preprocessing without smoothing, 3 s peak width, baseline subtraction just above the noise level, and automatic mass spectral deconvolution and peak detection at signal/noise levels of 5:1

throughout the chromatogram. Apex masses are reported for use in the BinBase algorithm. Result.txt files are exported to a data server with absolute spectra intensities and further processed by a filtering algorithm implemented in the metabolomics BinBase database. Raw results data need to be normalized to reduce the impact of between-series drifts of instrument sensitivity, caused by machine maintenance, aging and tuning parameters. There are many different types of normalizations in the scientific literature. We did a variant of a 'vector normalization' in which we calculated the sum of all peak heights for each sample's identified metabolites (but not the unknowns!). We call such peak-sums "mTIC" in analogy to the term TIC used in mass spectrometry (for 'total ion chromatogram'), but with the notification "mTIC" to indicate that we only use genuine metabolites (identified compounds) in order to avoid using potential non-biological artifacts for the biological normalizations, such as column bleed, plasticizers or other contaminants. Subsequently, we determined if the mTIC averages are significantly different between treatment groups or cohorts. If these averages are different by $p < 0.05$, data will be normalized to the average mTIC of each group. If averages between treatment groups or cohorts are not different or treatment relations to groups are kept blinded, data will be normalized to the total average mTIC. Both the processed and raw data files are uploaded to the database Metabolomics Workbench with StudyID ST002419.

### Benchmarking methods

We compared the performance of our Decrem with other five methods that are utilized to flux prediction and analysis: FBA, pFBA, FVA, REPPS and RELATCH, The cobra 2.0.5 package is utilized to implement the FBA, pFBA and FVA analysis, REPPS package is download on the address:https://academic.oup.com/bioinformatics/article/33/6/893/2725488?searchresult=1#supplementary-data, and the RELATCH can be found in https://genomebiology.biomedcentral.com/articles/10.1186/gb-2012-13-9-r78#MOESM12 (Additional File 12: Implementation of RELATCH. RELATCH is implemented using the COBRA Toolbox for MATLAB. (ZIP 173 KB)).

### Statistics and reproducibility

In the experiments, we used the complete samples from the datasets, without using any statistical methods to select or remove samples. Statistical significance was evaluated using Student's $t$-test or one-way ANOVA for parametric data and Wilcoxon rank-sum test or one-sided $t$-test for non-parametric data. The statistical analyses were performed using MATLAB R2020a. All the experiments can be reproduced by using the data and code that we uploaded to the public repository.

### Reporting summary

Further information on research design is available in the Nature Portfolio Reporting Summary linked to this article.

## Data availability

All data used are publicly available. The original and reconstructed metabolic models are available online: original metabolic models are available at BIGG models http://bigg.ucsd.edu/models/iND750, http://bigg.ucsd.edu/models/iMM904, http://bigg.ucsd.edu/models/iML1515, http://bigg.ucsd.edu/models/iAF1260 and reconstructed metabolic models of the four reconstructed models, iAF1260, iML1515, iMM904, and iDN750, are available at https://github.com/lgyzngc/Decrem-1.0/tree/master/three%20reconstructed%20models. All used exchange reactions, nutrient uptake, experimental growth rates, 13C fluxes and gene expression for Decrem modeling and metabolic simulation are found in Supplementary Data files. And the LS-MS data is sourced from https://www.ebi.ac.uk/biostudies/studies/S-BSST5?query=S-BSST5 for genome-scale mutant strains of E. coli. The RNA-seq data generated in this study have been deposited in the NCBI SRA database under accession code PRJNA910919. The metabolome data

are available at Metabolomics Workbench with StudyID ST002419, and the LS-MS data are available in the public Zenodo repository (https://doi.org/10.5281/zenodo.8285915)[73]. ALL data acquired in this study are also available in the public Zenodo repository (https://doi.org/10.5281/zenodo.8285915)[73]. Source data are provided with this paper.

## Code availability

Decrem is implemented as a MATLAB R2020a package. The source code, user tutorial and demo are available at GitHub (https://github.com/lgyzngc/Decrem-1.0.git) and Zenodo (https://doi.org/10.5281/zenodo.8285915)[73].

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

## Acknowledgements

The authors thank Schellenberger J's open-source MATLAB package: Cobra, and Saa PA's open-source MATLAB package: Fast_SNP for the integration of Decrem. This work was supported by the National Natural Science Foundation of China 62202334 (G.L.), 32171565 (H.C.), 61872418 (W.D.) and 62372494 (W.D.), Duke Kunshan Foundation (Chancellor's Fund) (H.C.), Kunshan Government Research Fund (H.C.), and Wang-Cai Foundation Grant (H.C.).

## Author contributions

G.L. and H.C. conceived and designed the work. G.L. and W.D. carried out computer implementation and data analysis. L.L. conducted the wet lab work. G.L., H.C., and W.D. interpreted the simulation results. G.L. and H.C. wrote the original manuscript, and W.D., and H.C. contributed to the writing of the final manuscript. All authors reviewed the final version of the manuscript.

## Competing interests

The authors declare no competing interests.
