## [Peer Review File · Nature Communications]

Reviewers' Comments:

Reviewer #1:

Remarks to the Author:

Abstract is quite vague and does not provide any meaningful information. For example it is not clear what methods were used by the authors and also what the improvement. In predictions are compared to existing models. Also suddenly Decrem shows up and we as readers have no idea of what this is. This is a really poorly written abstract.

Authors have tried to cram too many things into this paper. Decrem, pDecrem and kDecrem kpDecrem without explaining the differences with the result that this comes across as a very poorly written paper. Authors should indicate what these differences are why they need to invent so many methods. What is wrong with decrem, pdecrem that they have to invent kdecrem and kp decrem not to mention what they are doing in these different methods....
How does the kdecrem and kpdecrem compare with the kinetic models from the maranas group ? It does not seem fair to compare these with the FBA pFBA etc which are Stoichiometric models...

Figure 5 What is the point of comparing the growth rate to the PC vectors. !!! I would rather like to see the comparison with experimental values like in Figure 3. I am not sure I am convinced that the Minor improvements in prediction quality are worth it but the WT predictions. I would like to see more comparison with the mutants and RELATCH. RELATCH paper has some nice mutant data that can be compared.

Figure 6 showing the gene expression comparison is interesting. But I assume that these "predicted" gene expression are derived from fluxes and so it is important to indicate how this is calculated. This detail is not presented clearly in the paper. Even here why cannot all metabolic genes be compared with flux derived inferred gene expression. Why cherry pick a subset ? It seems that the gene expression data is correlated to the fluxes using a PLS model which means this is almost fit as opposed to a true prediction....

Step 1 in methods. How does this method compare with other methods that look at coupling such as the flux coupling finder or the Fast flux coupling calculator (F2C2) or the one used in FLFS-FC, none of which are cited here...

It might be useful to demonstrate this algorithm on a small sample metabolic network so that it is easier for the readers to understand the method. Some of the information on pDecrem and Decrem is buried in the SI and should be moved into the main MS with an example to show how this can be different than FBA

Line 600-608 It is not clear how this linear optimization will be different from FBA. I understand it might have fewer variables given that the coupled reactions are treated as one. In theory Decrem results and FBA Should be identical right ?

Minor concerns:

Line 22 mathematic to be changed to mathematical

Figure 3 I am not sure what B. subtilis is ? Could this be B.subtilis

Line 613

What are the input and output variables. One is presumably the gene expression and is the other metabolite concentrations ???

Where is equation 21 coming from ? How do they get these kinetics

Reviewer #2:

Remarks to the Author:

Li and co-workers present "Decrem," a network reduction method for improving genome-scale model prediction by identifying coupled reactions and reconstructing the metabolic network based on sparse linear basis reaction sets. The authors also incorporate gene regulation by activating/inhibiting metabolites into the kinetic modeling workflow to better constrain reaction flux at key metabolic branchpoints. The authors claim superior performance over existing constraint-based modeling frameworks and demonstrate application to three model microorganisms (*E. coli*, *B. subtilis*, and *S. cerevisiae*). The authors offer a number of statistical tests and comparisons as proof of their method's superior performance.

The authors presentation of their workflow, the details of their implementation, and how the different subparts are combined into a single modeling framework is unclear. The primary contribution of the work is the Decrem algorithm, and the novel application of the MinSpan algorithm, but the details of the Decrem algorithm are confined to the supplementary methods, with details of how to identify reaction cycle subnetworks, incorporate them into a single flux balance analysis problem, find correlations between metabolite concentration and gene expression, and linearize kinetic rate expressions in the primary methods text. A major portion of the contribution of this work is also the incorporation of kinetics into the Decrem framework. However, the details of the kinetic portion of the workflow is disjointed and incoherent in the main body of the manuscript. The source codes for Decrem makes extensive use of pre-existing packages, and are thus not generalized for application to models outside of the three examples given. The impact of this work would benefit greatly from a generalized package with clearly defined data inputs.

Major Concerns

How does the sparse basis vectors-based network reduction in Decrem compare to previous methods, such as Kron reduction (Rao et al., 2014, BMC Sys Bio), Metatool (Pfeiffer et al., 1999, Bioinf), minimal reaction sets (Burgard et al., 2001, Biotech Prog), NetworkReducer (Erdrich et al., 2015, BMC Sys Bio), and redGEM (Ataman et al., 2017, PloS Comp Bio)? What is the advantage of using the method for identifying coupled reactions deployed in this study over an already established method, such as flux coupling analysis for identifying coupled reactions in the metabolic network?

Why was the iAF1260 GSM model used, instead of the latest iML1515? The latter has a higher reaction and gene content, which might affect the network decomposition reported in Decrem.

Line 29: The claim of "outstanding performance" is baseless and subjective without quantitative support. Consider simply stating that predictions were improved over existing methods.

Line 44: "in vivo" and "in a GSM" are two separate conditions. It should be distinguished whether the challenge in systems biology is understanding how reactions are regulated in vivo, or how this translates to GSM constraints. If both are challenges, then this should be restated clearly.

Line 47: "due to quick compensation from nearby reactions" requires a citation.

Line 48-49: The explanation needs to be amended as Schuetz et al. do not mention a "global optima", but rather suggest that microorganisms grow to satisfy multiple competing objectives, and flux adjustment upon genetic or environmental perturbation is minimized from a reference state.

Line 57: The authors state "recent studies", but provide a single citation.

Line 60: "interconnected with high kinetic capabilities" does not make sense.

Line 60: "small worlds" should be accompanied by a citation such as Wagner and Fell, 2001.

Line 66: The authors should clarify what is meant by "neighboring reactions".

Line 69: The citation provided does not support the accompanying statement.

Line 71: It is not clear what is meant by the statement that metabolic systems are "rather dynamic" in adjusting to perturbations. It seems obvious that introducing a perturbation to a system would result in a dynamic response by the organism. In the case of a genetic perturbation, the strain must be constructed before it is cultured. For this case, the statement is misleading if not entirely incorrect. Did the authors mean 'robust' instead of 'dynamic'?

Line 74-75: The BRENDA database contains a wealth of substrate-level regulatory interactions from reactions across genome-scale metabolism. Substrate-level regulations are particularly well documented for model organisms, such as those modeled in this study. This statement should be modified to reflect this.

Line 75-76: Transcriptional regulation of gene expression is well-documented for model organisms like E. coli for which information is consolidated in the regulonDB database. This statement should be updated to reflect this.

Line 80-82: It is unclear which if the regulatory metabolites referred to are transcription factor effector metabolites, or allosteric effectors of metabolic enzymes.

Line 89-93: Knowledge of key metabolites controlling metabolism is not useful without information about their pool sizes. This statement should be updated to reflect this and the need for metabolomics data in the k-Decrem framework.

Line 101: Is Decrem the name of the model, or the name of the method which is used to constrain the model?

Line 107-110: It is unclear from this sentence if metabolite concentrations are predicted by the k-Decrem, or are a data requirement for deploying k-Decrem (although mentioned later in Results).

Line 111: It is unclear what "genome-scale knockout strains" refers to – this sentence needs to be rephrased.

Line 111: It is unclear what is meant by "buffered" by highly coupled reactions", particularly if intracellular perturbations are genetic perturbations, as previously stated, a genetic intervention would likely have the same impact on a reaction coupled to the knocked out reaction. This statement should be clarified.

Line 143-145: It is not clear how single K_m values were determined for coupled and uncoupled reactions. There should be many K_m values, given the multiplicity of coupled reaction sets and the many uncoupled reactions. Statistical quantities (p-value) provided also require an explanation as to how they were implemented and why they were used.

Line 145: Any statement about physical proximity is unfounded, as the modeling framework in question does not take into account the spatial organization of cells.

Line 145-148: The physiological meaning of K_m is the substrate concentration at which the reaction rate is half of the maximum. This could indicate which reactions are likely operating at or close to V_{max} in a pathway, but there is no evidence that suggests reactions "interact" based on K_m values alone. Neither do cells prioritize metabolic function, nor are pathways capable of "reaching out" to each other.

Line 149: Citation 35, Bordbar et al., does not contain experimentally measured fluxes.

Line 151-154: It is not clear how uncoupled reaction sets were defined for the comparison in Figure 2C. Were reactions randomly sampled from the set of all uncoupled reactions, normalizing for the size of the coupled reaction sets?

Line 168-170: Because the emphasis of this work is on network topology and regulation, particularly in the case of *E. coli*, the authors should see if they can establish a correlation between SLBs and specific operons, this would provide more concrete evidence that their workflow is capturing biology, and not just an artifact of the decoupling framework.

Figure 2E: why do low reaction fluxes deviate most in the decomposed vs original model? Do these low fluxes necessarily constitute trivial reactions, as their numerical values are determined by factors such as the coefficient of the end metabolite (of a given pathway) in the sink reaction (such as biomass). Can other basis of comparisons be also reported, such as net ATP/biomass/cofactor yield?

Line 185: "Decrem" is defined here as a model, but three models are constructed. It needs to be made clear whether "Decrem" is a model or if it is the workflow for reconstructing/constraining the model.

Line 188-189, 194: If the Decrem FVA solution space preserves the original GSM model FVA solution space, then it is unclear what advantage Decrem offers over standard FBA or FVA – does Decrem help reduce the original FVA flux bounds, thereby reducing uncertainty? No citation should be given after this sentence if it is truly a result of the modeling framework that is proposed here. Also (Line 194), FBA predicts a single feasible point within a solution space. To demonstrate improved Decrem predictive capability, the authors should demonstrate how the FVA ranges predicted by Decrem shrink compared to FVA ranges predicted with the original genome-scale model while still capturing the 95% confidence intervals estimated using 13C-metabolic flux analysis.

Line 190: It is unclear what constitutes a "trivial flux", and what constitutes a "primary flux". This should be clarified.

Line 193: This statement makes it seem as if Decrem is a single model which is used to predict phenotype for each of the three listed organisms. It should be made clear that the Decrem workflow was used to reconstruct GSMs for the three mentioned organisms.

Line 208: To demonstrate activity, it should be demonstrated that the lower bounds for the Decrem-predicted flux distribution is non-zero. To truly demonstrate agreement with 13C-MFA-estimated flux, 13C-MFA should be carried out with a genome-scale atom mapping model, and it should be demonstrated that the lower bound of the reactions in question are non-zero when all alternative atom mappings are present in the atom mapping model.

Line 210-212: FVA ranges predicted by Decrem should be compared to those predicted by the original stoichiometric model and those estimated with 13C-MFA to demonstrate quantitative agreement between the Decrem-predicted flux ranges and 13C-MFA-estimated flux ranges.

Figure 3C: why were only TCA cycle reactions selected for the comparison? How are the reported performance metrics affected when the entire network (if experimental fluxes are available), or rest of the core metabolism is considered?

Line 226-227: The authors mention "cellular compartments", but the cited source for *S. cerevisiae* 13C-MFA fluxes also does not distinguish between compartments in their core metabolic model. It is, therefore, not clear as to how compartment-wise flux allocation was determined and compared to Decrem predictions.

Line 228-252: It is poor practice to compare GSM-predicted fluxes from a single solution within the solution space to 13C-MFA-predicted fluxes. Instead, FVA ranges predicted by each modeling framework should be compared to 13C-MFA-estimated confidence intervals. The lack of compartmentalization in yeast atom mapping models also needs to be accounted for when

comparing GSM predictions.

L228-277: It is not clear how cases with gene knockouts in reactions belonging to a coupled set (hence merged into a linear basis reaction (LBR)) were handled. Was flux through the LBR forced to be zero or were coupled reaction sets recomputed, leaving out the affected reaction? Similarly, can the kinetic parameters estimated for a LBR be interpreted physiologically, perhaps corresponding to an 'effective' turnover rate?

Line 260: The PCA acronym is never defined in the text.

Line 264-267: Can the authors provide clarification as to why the maximum growth rate predicted by Decrem is not identical to that predicted by FBA with the original model if both maximize growth rate? Are there growth-coupled reactions which are being removed from the Decrem network?

Line 267-269: Did the other methods predict similar growth rates for the mentioned strains?

Line 267-269: If Decrem was not used to predict the genetic mutations leading to low growth rate, then please rephrase to state that Decrem was able to predict a low growth rate in those strains, rather than identify the strains.

Line 275: "Large-effect mutations" is un-descriptive, and the authors make the claim that other constraint-based methods lead to a high false-positive rate when assessing growth rate. The authors should clarify what is meant by "false-positives", and provide specific examples where Decrem is able to better predict growth phenotypes than other constraint-based methods (the ZWF1 example in the text fails to discuss predictions made by the other methods).

Line 290-291: Particularly for E. coli, for which transcriptional regulatory information is consolidated in a single database, there is a wealth of information linking effector metabolites with transcription factors and transcription factors with regulated enzymes. It should be made clear how the developed framework both supports the existence of these documented effector metabolites and expands on the number of potential effector metabolites.

Lines 289-356: No mechanistic basis for the transcription model developed in this study is presented. The Michaelis-Menten formalism describes allosteric interactions with the enzyme itself. The authors should provide a mechanistic meaning for the developed model which links the kinetics of transcription to enzyme kinetics if the intent is to describe transcriptional regulation using enzyme kinetics.

Lines 336-341: The observation that the expression of growth-associated pathways correlate to the concentration of biomass components is expected. It would be more interesting if the authors could relate gene expression to other effector metabolites that interact with transcription factors such as cAMP, and relate those interactions back to transcriptional regulations documented in regulonDB (in the case of E. coli).

Line 348-356: If the kinetic model is predicting fluxes for the same conditions that it was fitted to, then the model is not predictive and merely fits the data well. Please clarify whether the model is being used to predict phenotype or if it is only being fitted to data. Details about the kinetic model itself are also absent and should be provided.

Line 361-380: It is presumable the added kinetic constraints caused a decrease in the growth rate of the knockout strains. Can the authors confirm that the improved correlation was due to a decrease in the predicted growth rates?

Line 382: Please define the term "topological vulnerability".

Line 395-399: A discussion of the Relatch and REPPS methods is also warranted, as their predictive accuracy is the same as Decrem and k-Decrem, if not higher.

Figure 8: A comparison with kinetic models of *E. coli* metabolism (such as Khodayari and Maranas, 2016, Nat. Comm., Millard et al., 2017, PLoS Comp Biol., Kurata et al., 2018, J Biosci Bioeng., Gopalakrishnan et al., 2020, Metab Eng.) is also necessary, especially as Decrem is an optimization-based framework and uses biomass as the objective function, which has been shown to not be valid under every growth conditions.

Line 453-456: Is Decrem a model or a modeling framework? Throughout the manuscript, it is compared alongside formalisms such as FBA, pFBA, RELATCH, etc. This needs to be made clear.

Line 469-470: The authors should remove "in biochemistry textbooks" and provide citations to support their statement. If a citation cannot be found to support the statement, then it should be removed. Also, the authors are recommended to use regulonDB for obtaining genome-wide transcriptional regulatory information in *E. coli*. Please update the statement to reflect this knowledgebase or remove it.

Line 505: It is not clear what is meant by "coactivated" reaction cycle. The TCA cycle is given as an example of a "coupled, coactivated reaction cycle".

Line 525, 528, 530: How the equations are used in the context of the Decrem workflow is never stated. This should be made clear in the methods section, perhaps by including the workflow chart given in Supplementary Figure 1.

Line 534: Throughout the manuscript, "highly coupled" reactions and substructures are mentioned, but what defines a "highly coupled" set of reactions is never fully defined. The authors should define what constitutes a "highly coupled" reaction.

Line 553: The formulation needs to be restated to be more easily understood. There is also no way to tell what the purpose of Step 2 is in the broader workflow without referring to the supplementary materials. If the authors directly applied the MinSpan algorithm in this step, they should state this with the citation and avoid math notation. If the authors somehow adapted the MinSpan algorithm for a novel use, they should state this and how their formulation differs from the original algorithm.

Line 576: Can the authors elaborate on what the purpose of this optimization formulation is in the context of the overall Decrem workflow?

Line 603: Can the authors provide a mechanistic basis for the expressions used to establish transcriptional relationships in this section? Methods such as that used by Kochanowski et al. to identify regulatory metabolites coordinating gene expression have attached biophysical meaning to the expressions they have used to probe for effector metabolites.

Line 603: From reading the methods in the main body of the manuscript, it is unclear how this section ties into the other methods in the workflow. Can the authors link this section to the other sections within the methods section, rather than only doing so in the supplementary methods?

Line 686: SIMMER, as described in the accompanying citation, uses convenience kinetic rate laws. Hence, it is unclear why the authors use a logarithmic transformed Michaelis-Menten rate expression for this section.

Line 686: If SIMMER was used to assess the ability of hypothetical regulations to improve fitness when assembling rate expressions and estimating kinetic parameters, and if regulations not found in literature were included in the kinetic model, then statistical analysis of results for all regulations tested should be included in the manuscript or supplementary files, to provide reasoning for why specific rate expression and parameter values were chosen.

Line 692-706: This section does not adequately describe how the kinetic formalisms described in the previous two sections are incorporated into the FBA formulation. The authors should re-write this sections to clearly explain how a steady-state flux distribution is constrained using predictions from the kinetic model and transcriptional model detailed in the previous sections, with attention

paid to the manner in which Equation 32 is formulated and described.

Supplementary files:

Materials S3: The authors mention that they have used the transcriptional regulations identified to inform parameterization of a Michaelis-Menten model describing the flux through enzymatic reaction in the metabolic network. Is there a mechanistic basis or historical precedent that allows the authors to express a transcriptional regulation (i.e., data collected from citations 8-10) as an allosteric regulation? Expressing a transcriptional event in this manner is problematic because the mathematical representation of allosteric regulation within the Michaelis-Menten formalism has mechanistic meaning.

Minor Concerns

Line 168: "gene level" should be "gene expression level".

Line 206: "Empirical fluxes" should be "Experimental fluxes"

Section title 'Building a metabolic kinetic model incorporating metabolite-TF regulation' is misleading, as TFs are not explicitly modeled. Recommend changing it to emphasize that regulatory metabolites are modeled.

Line 361: repetition of the phrase 'growth rates'

Line 379: "Empirical" should be "Experimental"

An overall response to both reviewers:

The comments made by two reviewers are very constructive and instrumental in enabling us to enhance this work and rewrite the manuscript. For that, we wholeheartedly extend our gratitude to both of you. We emphasize that this revised manuscript is almost rewritten by adding new wet-lab work and more computational analyses, which are highlighted in green in the main text.

With the comments from reviewers, we made the following changes:

- Completely redesigned and executed the research and rewrote the manuscript;
- Conducted wet-lab experiments with *E. coli* to collect transcriptomes and metabolomes in MOPS minimal medium at four timepoints along the growth curve (see Methods in the revision);
- Performed new analyses as required or suggested by reviewers, e.g., using the latest *E. coli* GSM model iML1515 which produced new figures, flux variability analysis, developing a new global-regulation-based kinetic mechanism, and introducing large test datasets;
- A new transcriptional regulation mechanism was developed to fit and explain the observed linear relationship between the metabolites and gene expression (derived from a previous study [1]), which also derived the linear kinetics for the global cell state-regulated reactions. All the reformulated mechanisms were validated on the independent test datasets from our wet experiment or extra data source.
- All original figures were modified or removed and new ones were drawn;
- The tool has also been rewritten in Matlab and made available at <https://github.com/lgyzngc/Decrem-1.0>.

Reviewer 1:

Abstract is quite vague and does not provide any meaningful information. For example it is not clear what methods were used by the authors and also what the improvement. In predictions are compared to existing models. Also suddenly Decrem shows up and we as readers have no idea of what this is. This is a really poorly written abstract.

Response: We rewrote the Abstract in the revised manuscript addressing the comments above highlighted the key idea and biological mechanism of Decrem, and explained its main improvements compared to existing models.

Authors have tried to cram too many things into this paper. Decrem, pDecrem and kDecrem without explaining the differences with the result that this comes across as a very poorly written paper. Authors should indicate what these differences are why they need to invent so many methods. What is wrong with decrem, pdecrem that they have to invent kdecrem and kp decrem not to mention what they are doing in these different methods...

Response: Thanks for pointing out. Initially, we were indeed trying to make a comprehensive case highlighting our model, method, and biology. With the reviewers' comments, we decided to redesign the research and focus on the method for reconstructing and constraining GSM

models with local topological coordination and global cell state regulation. In the revision, we only have one variant version of Decrem, pDecrem (parsimonious Decrem, an equivalent of pFBA in which the optimization objective of parsimonious Decrem/FBA [2] minimizes the number of non-zero flux reactions while maintaining the given growth rates). So, Decrem and pDecrem are similar to FBA and pFBA, the two most popular methods of canonical metabolic flux analysis. (In the previous version, kDecrem means the Decrem constrained by the given kinetic fluxes (as is the FBA by extra fluxes); kpDecrem means kinetic flux-constrained pDecrem). In the revised version, we removed the naming of kDecrem and kpDecrem for simplicity.

How does the kdecrem and kpdecrem compare with the kinetic models from the Maranas group ? It does not seem fair to compare these with the FBA pFBA etc which are Stoichiometric models...

Response: The kinetic model from Maranas group is k-ecoli457 (A genome-scale *Escherichia coli* kinetic metabolic model k-ecoli457 satisfying flux data for multiple mutants trains: <https://doi.org/10.1038/ncomms13806>). In Decrem, we first predicted the kinetic fluxes for those cell state (i.e., growth rate)-regulated reactions, and then constrained the GSMs or the decoupled metabolic model from Decrem with those kinetic fluxes. This is a two-phase process and all the methods compared (including FBA and pFBA) are carried out under the constraint of identified kinetic fluxes.

For Maranas's kinetic model, they infer the reaction kinetic using an elementary reaction-based metabolic network. The primary difference between Decrem and Maranas's kinetic model include: (1) Decrem estimates the reaction kinetics, which is primarily regulated by global growth state, with several experimental ^{13}C fluxes and identified regulatory metabolite concentration, and is used to constrain canonical GSM. In contrast, Maranas's kinetic model is a rebuilt median-scale kinetic model and utilizes the genome-scale experimental ^{13}C fluxes and metabolite concentration data. However, limited by the current mass spectra and ^{13}C trace techniques, only few hundreds metabolites and less than 100 reactions can be identified experimentally, so the training data used in Maranas's kinetic model is partly generated in a simulated way, which causes the high complexity of kinetic modeling and may introduce extra noise into kinetic model. (2) Decrem can be expanded to new perturbed strains using the well-constructed GSM models (e.g., *i*ML1515 or *i*AF1560) with the experimental metabolite concentration data, and is much flexible. In contrast, Maranas's kinetic model shrink reactions and metabolites using the decomposed elementary reactions, which is function constrained and cannot be expanded. Genome-scale ^{13}C flux data is also needed for new perturbed strains in Maranas's kinetic model. Nevertheless, for single-gene deletion strains of *E. coli* with ^{13}C flux data, we compared the predicted fluxes using these two kinetic methods and two other kinetic models: kurata and k-FIT and found the results are quite similar between them (Table R1). Note that our model achieves comparable accuracy using only five global regulated-reaction kinetics, which primarily relieves the data and computational cost of building the reaction kinetics.

Table R1. Comparison of consistency between predicted fluxes and measured ^{13}C fluxes

for four test methods.

Strain	kurata	k-ecoli457	k-FIT	Decrem
glk	0.76	0.92	0.99	0.89
pgi	0.77	0.93	0.99	0.91
pfka	0.75	0.96	0.99	0.87
pfkb	0.8	0.96	0.97	0.9
fbp	0.83	0.95	0.99	0.89
fbab	0.82	0.98	0.98	0.91
gpma	-0.22	0.99	0.99	0.87
pyka	0.83	0.98	0.98	0.91
pykf	0.89	0.94	0.95	0.87
ppsa	0.78	0.99	0.93	0.89
zwf	0.9	0.95	0.99	0.91
pgl	0.2	0.99	0.98	0.85
gnd	0.24	0.97	0.99	0.89
rpe	0.16	0.99	0.97	0.9
rpia	0.83	0.94	0.95	0.9
rpib	0.8	0.95	0.98	0.91
tkta	0.72	0.95	0.96	0.89
tktb	0.74	0.99	0.99	0.89
tala	0.76	0.99	0.99	0.92
talb	0.76	0.99	0.98	0.9
ppc	0.79	0.93	0.99	0.89
wt 0.1	0.95	0.96	0.97	0.93
wt 0.4	0.94	0.81	0.98	0.88
wt 0.5	0.9	0.95	0.97	0.92
wt 0.7	0.86	0.97	0.98	0.9

Figure 5 What is the point of comparing the growth rate to the PC vectors. !!! I would rather like to see the comparison with experimental values like in. Figure 3. I am not sure I am convinced that the Minor improvements in prediction quality are worth it but the WT predictions. I would like to see more comparison with the mutants and RELATCH. RELATCH paper has some nice mutant data that can be compared.

Response: We used the top two PC vectors to represent the growth rate-derived variance for the yeast mutant strains in our analysis for two reasons: (1) Kochanowski et al. found that the growth dominates the first PC of gene expression in central metabolism rate-derived global regulation constraint [3]; (2) Sánchez et al [4] found that the top two PCs of flux distributions in metabolic models can represent the growth rate in Yesat mutant strains (cell state, which is different from the biomass reaction). Based on these two points above, we characterized the growth rate for multiple mutant yeast strains using two top PCs of flux distribution. Following the reviewer's suggestion, we included 38 yeast mutant strains for comparison in MSE, flux distribution, and activated reactions and six *E. coli* mutant strains (which have proven hard to

predict), among the methods FBA, pFBA, REPPS, RELATCH, and Decrem (Fig. R1; below), which is also used in the RELATCH paper.

Fig. R1. Comparison in performance among FBA, pFBA, Decrem, and Relatch on six *E. coli* mutant strains using the iML1515 model and 38 yeast mutant strains using the iMM904 model.

Figure 6 showing the gene expression comparison is interesting. But I assume that these "predicted" gene expression are derived from fluxes and so it is important to indicate how this is calculated. This detail is not presented clearly in the paper. Even here why cannot all metabolic genes be compared with flux derived inferred gene expression. Why cherrypick a subset? It seems that the gene expression data is correlated to the fluxes using aPLS model which means this is almost fit as opposed to

a true prediction...

Response: First, we would like to point out that we predict the gene expression with metabolite concentrations rather than ^{13}C fluxes. Following your suggestion, in the revision we provided the biological mechanism foundation for metabolic expression prediction, by citing the paper entitled "*Few regulatory metabolites coordinate expression of central metabolic genes in Escherichia coli*". We also developed a mechanistic model to link the transcription kinetics to metabolite regulators based on the gene regulatory model [3]. From this model, we can get a linear regulation model between the central metabolic gene expression, local regulators, and the global regulators.

$$\log(E_g) \approx \alpha_g \log(R) + \sum_{i=1}^K \beta_{gi} \log(M_{gi}) + c$$

where the E_g represents the expression E of gene g , R indicates the given growth rate, and M_{gi} represents the i th of K metabolite regulators, then α_g and β_{gi} represent the corresponding coefficient. According to the kinetic of biomass reactions, we can represent the growth rate as:

$$R = \lambda \prod_{j=1}^N (1 + M_{bj}/K_{mj})^{\theta_j}$$

where the M_{bj} represents the j th of N biomass metabolites, and K_{mj} is a cell state-related kinetic parameter, λ and θ_j are the reaction coefficients, following Sauer's paper[3], we approximate $\log(1 + M_{bj}/K_{mj})$ with $\log(M_{bj}/K_{mj})$. We take logarithm operation for above equation and approximate it as:

$$\begin{aligned} \log(R) &= \sum_{j=1}^N \theta_j \log(1 + M_{bj}/K_{mj}) + \log(\lambda) \\ &\approx \sum_{j=1}^N \theta_j \log\left(\frac{M_{bj}}{K_{mj}}\right) + \log(\lambda) \\ &= \sum_{j=1}^N \theta_j \log(M_{bj}) + c \end{aligned}$$

So,

$$\log(E_g) \approx \alpha_g \sum_{j=1}^N \theta_j \log(M_{bj}) + \sum_{i=1}^K \beta_{gi} \log(M_{gi}) + b$$

where M_{bj} indicates the biomass metabolites and M_{gi} represents the TF regulatory metabolites, we get an approximate quantitative relationship between gene expression and metabolite concentration in the central metabolism.

Depending on the potential linear relationship between the normalized metabolic regulators and gene expression above in central metabolism of *E. coli*, we first tested the correlation between 85 genes expression and concentrations of 45 candidate regulatory metabolites in central metabolism of 24 *E. coli* mutant strains using the canonical correlation analysis (CCA), poor correlation scores between their top two canonical variables (representing 90% variance)

were obtained (Fig. R2):

Fig. R2. The CCA correlation between the 45 metabolite concentrations and 85 gene expression levels in central metabolism of *E. coli*.

Next, we clustered the correlation between the concentrations of 45 metabolites and 85 gene expression. Then candidate regulator metabolites can be clustered into the biomass and regulator group (see revised manuscript for details). We then tested the correlation between the concentration of metabolites in either each group with the 85 gene expressions and found that all canonical variables show correlation coefficients greater than 0.8 in the biomass group whereas no significant correlations were observed in the regulator group (Figs. R3 and R4). These observations are consistent with the established knowledge: central metabolism genes are primarily regulated by the general global cell state and enhanced by the specific regulators [3]. This result explains the linear relationship between the biomass composition and gene expression in central metabolism.

To identify the genes which are only determined by the global cell state, we used a partial least squares regression (PLSR) to fit the 85 gene expressions with biomass group metabolites with a leave-one-out cross-validation. We filtered the genes with a cutoff of regression coefficient 0.84. The first PC (accounting for >38% variance) is the global cell state-derived genes, 32 genes are selected from the total of 85 genes in central metabolism and find most of them are located in PPP pathway. We then calculated the correlation between the identified genes and biomass metabolites and obtained a correlation coefficient >0.9 for each canonical variable (Fig. R5).

Fig. R3. The CCA correlation between the biomass composition metabolites and 85 genes in central metabolism of *E. coli*.

Fig. R4. The CCA correlation between the specific regulator metabolites and 85 genes in central metabolism of *E. coli*.

Fig. R5. The CCA correlation between the biomass metabolites and identified 32 genes in central metabolism of *E. coli*.

We then test the identified 32 genes and biomass group metabolites on our multi-omics experimental dataset (see manuscript for details) and also find a strong canonical correlation score (Fig. R6)

Fig. R6. The CCA correlation between the biomass metabolites and identified 32 genes in central metabolism of *E. coli* in a time series dataset.

Finally, to estimate the statistical significance of the selected biomass metabolites, we performed random sampling 10,000 times for the 45 metabolites and estimated the regression

coefficient for 32 genes with those sampled metabolites. The P value is 0.0031 for biomass metabolites vs 0.48 for regulator metabolites, which validates the identified relationship between the gene expression and biomass metabolites.

Step 1 in methods. How does this method compare with other methods that look at coupling such as the flux coupling finder or the Fast flux coupling calculator (F2C2) or the one used in FLFS-FC, none of which are cited here...

Response: Thanks for this comment. We cited the three papers mentioned above in our revised manuscript and explained their two major differences from Decrem: (1) Decrem identifies the coupling reactions based on the network topology. Reactions are considered as coupled if they are located in a cycle of reaction set, which all start with the same substrates and end with the same products (sourced from and sink to the same metabolite). Reaction coupling will lead to a coupled-reaction structure at the sub-network level based on the topological property of metabolic networks (e.g., small world and high clustering coefficient). In contrast, the Flux Coupling Finder (FCF) and Fast flux coupling calculator (F2C2) deem reactions as coupled reactions if two reactions are activated simultaneously, which are based on mathematically coupled fluxes, and FLFS-FC uses the Fast flux coupling to reduce the computational cost of EFM enumeration. (2) Coupled-reaction subnetworks identified by Decrem are represented as a group of minimal sparse basis reactions for metabolic network reconstruction, which is similar to the conception of the minimal pathway. However, the coupled reactions are inseparable in the three methods cited above.

It might be useful to demonstrate this algorithm on a small sample metabolic network so that it is easier for the readers to understand the method. Some of the information on pDecrem and Decrem is buried in the SI and should be moved into the main MS with an example to show how this can be different than FBA

Response: Thanks for this very constructive suggestion! We did as suggested in the revised manuscript: we have included a demonstration of the sample metabolic network in the schematic chart and showed how coupled reactions are decoupled (Fig. 1 in the main text). Additionally, we have provided the details of Decrem and pDecrem (parsimonious Decrem in revision) in the revision. Both are used to show their differences from FBA.

Line 600-608 It is not clear how this linear optimization will be different from FBA. I understand it might have fewer variables given that the coupled reactions are treated as one. In theory Decrem results and FBA Should be identical right ?

Response: The short answer is that the results should differ, but the general optimization is similar. By treating each sparse basis of couple reactions into one, we reconstructed a topologically decoupled metabolic network, which constrains the flux passing the sparse basis component reactions with the same scale value. In other words, we introduce extra flux coordinating constraints for the coupled reactions within the same sparse basis, which will reconstruct the canonical GSM, so the optimal space of Decrem is shrunk by coordinating constraints within the sparse linear basis. Given this significant difference, the results from Decrem are better than FBA as shown in the text, although the general optimization is similar.

Minor concerns:

Line 22 mathematic to be changed to mathematical

Response: this wording is no longer used in the revised manuscript.

Figure 3 I am not sure what B. bubtilis is ? Could this be B.subtilis

Response: Yes, it is B. subtilis, so we changed it to the Gram-positive bacterium *Bacillus subtilis* in the revised manuscript.

Line 613

What are the input and output variables. One is presumably the gene expression and is the other metabolite concentrations ???

Response: we can consider the partial least squares regression as a canonical correlation analysis plus a multiple linear regression, specifically, for $X = TP^T + E$, the X represents gene expression matrix, T represents the dimension reduced principal component matrix and P represents corresponding loading matrix, in addition, T are also constrained to maximize the similarity with U ($Y = UQ^T + F$) where U represents the principal component matrix of metabolite concentration Y . So the input of line 767 is X , the gene expression matrix, and output are T and P , the dimension reduced component matrix and its loading matrix, as well as an error matrix E

Where is equation 21 coming from ? How do they get these kinetics

Response: the equation 21 in first version (now Equation 25 in the revised manuscript) was cited from the formula 11 of Chapter Two of the "Enzyme kinetics" paper (Segel I H. Enzyme kinetics: behavior and analysis of rapid equilibrium and steady state enzyme systems[J]. 1975.). They assumed an steady-state of enzyme catalysis of the reverse reaction, and then got the total reaction rate after a series of complex derivation. Based on the formulation 11 in Segel I H's paper, we also appended the activator and inhibitor kinetic to the basic reverse kinetic to simulate the realistic intracellular reactions.

Reviewer 2:

The authors presentation of their workflow, the details of their implementation, and how the different subparts are combined into a single modeling framework is unclear. The primary contribution of the work is the Decrem algorithm, and the novel application of the MinSpan algorithm, but the details of the Decrem algorithm are confined to the supplementary methods, with details of how to identify reaction cycle subnetworks, incorporate them into a single flux balance analysis problem, find correlations between metabolite concentration and gene expression, and linearize kinetic rate expressions in the primary methods text. A major portion of the contribution of this work is also the incorporation of kinetics into the Decrem framework. However, the details of the kinetic portion of the workflow is disjointed and incoherent in the main body of the manuscript. The source codes for Decrem makes extensive use of pre-existing packages, and are thus not generalized for application to models outside of the three

examples given. The impact of this work would benefit greatly from a generalized package with clearly defined data inputs.

Response: we appreciate this very constructive suggestion. Following it, we reorganized the manuscript to make clear the workflow and details of algorithm implementation of Decrem algorithm and to explain the different subcomponents using a schematic chart and how the kinetic portion is incorporated into Decrem (Fig. 1 in the main text, which is provided below). We explain our Decrem as the reconstructed GSM using the reformulation of local coordinated reactions and global regulated reactions. In addition, we rewrote the software package for generalized usage with a detailed user demo.

Figure 1. The schematic chart of Decrem

Major Concerns How does the sparse basis vectors-based network reduction in Decrem compare to previous methods, such as Kron reduction (Rao et al., 2014, BMC Sys Bio), Metatool (Pfeiffer et al., 1999, Bioinf), minimal reaction sets (Burgard et al., 2001, Biotech Prog), NetworkReducer (Erdrich et al., 2015, BMC Sys Bio), and redGEM (Ataman et al., 2017, PloS Comp Bio)? What is the advantage of using the method for identifying coupled reactions deployed in this study over an already established method, such as flux coupling analysis for identifying coupled reactions in the metabolic network?

Response: we explained the key differences between Decrem and the methods mentioned above in the Introduction of the revised manuscript. First, Decrem coordinates the coupled fluxes with the sparse linear basis vectors (SLBs) from the identified coupled sub-network to reconstruct a topologically decoupled metabolic model. At the same time, the purposes of the methods mentioned above are to reduce the redundant reactions of metabolic networks. Significantly, they are often based on single reactions rather than an entire network. Second,

the sparse basis-based reconstruction represents the local stoichiometric matrix of identified coupled sub-network in an orthogonalizing way, which is a reversible transformation and is only used to shrink the feasible region of corresponding coupled reactions. Similar to the conception of “Minimal metabolic pathway” [5], the identified coupled SLBs are topologically connected. At the same time, the Kron reduction represents the flux kinetics for a specific metabolite, which depends on the known kinetic parameters and is hard to use in a genome-scale network. For Metatools, it is similar to the flux control analysis, identifying paired reactions with a synchronous varying non-zero flux. A minimal reaction set is a strategy to find the minimal number of reactions loaded with non-zero fluxes under the given growth condition. NetworkReducer is similar to the minimal number of reactions with different constraints. redGEM has a similar searching strategy to reduce the metabolic network, but the input metabolites must be user predefined artificially. It is used to extract a specific pruned model from GSMs. The flux coupling analysis only works at the reaction level, i.e., paired reactions that coordinate their flux and aid the metabolic reconstruction. Overall, our Decrem identifies coupled sub-network based on topological property and decomposes this sub-network as sparse orthogonal basis to constrain the feasible region of coupled fluxes. In contrast, most of the above methods are based on paired reaction and they reduce the GSMs.

Why was the iAF1260 GSM model used, instead of the latest iML1515? The latter has a higher reaction and gene content, which might affect the network decomposition reported in Decrem.

Response: Thanks for this suggestion. Besides iAF1260, we also used the iML1515 model and based on it made metabolic flux predictions for six mutant strains of *E. coli*. Results are shown in the manuscript and presented here in Fig. R8 A, B and C below.

Fig. R8. The performance comparison of six mutant strains of *E. coli* are tested on the iML1515 model.

Line 29: The claim of “outstanding performance” is baseless and subjective without quantitative support. Consider simply stating that predictions were improved over existing methods.

Response: thanks for pointing this out: we replaced “outstanding performance” with “better predictions” in the revised manuscript.

Line 44: “in vivo” and “in a GSM” are two separate conditions. It should be distinguished whether the challenge in systems biology is understanding how reactions are regulated in vivo, or how this translates to GSM constraints. If both are challenges, then this should be restated clearly.

Response: we removed the *in vivo* from the revised manuscript.

Line 47: “due to quick compensation from nearby reactions” requires a citation.

Response: we added this citation: Fuhrer T, Zampieri M, Sevin DC, Sauer U, Zamboni N. Genomewide landscape of gene-metabolome associations in *Escherichia coli*. *Mol Syst Biol* 13, 907 (2017).

Line 48-49: The explanation needs to be amended as Schuetz et al. do not mention a “global optima” , but rather suggest that microorganisms grow to satisfy multiple competing objectives, and flux adjustment upon genetic or environmental perturbation is minimized from a reference state.

Response: Here, the “global optima” means the systems level and multi layers of metabolic adjustment, rather than the mathematical conception in linear programming.

Line 57: The authors state “recent studies” , but provide a single citation.

Response: we added another citation in the revision, that is, “Park JO, et al. Metabolite concentrations, fluxes and free energies imply efficient enzyme usage. *Nature Chemical Biology* 12, 482-489 (2016).”

Line 60: “interconnected with high kinetic capabilities” does not make sense.

Response: We meant it to be the preferred reaction interaction, For example, in the TCA (tricarboxylic acid) cycle, the reaction chain D-isocitrate  a-ketoglutarate  ...  malate ($K_m=0.029\mu\text{M}$ and $K_{cat}=106.4$ for IDH3 (Isocitrate dehydrogenase)) is preferred as the primary branch for oxidation of acetyl-CoA over the low-efficiency reaction chain D-isocitrate  succinate ($K_m =8\mu\text{M}$ and $K_{cat} =28.5$ for AceA (Isocitrate lyase)).

Line 60: “small worlds” should be accompanied by a citation such as Wagner and Fell,2001.

Response: thanks for pointing this out, we added several citations for “small worlds” in the revised text.

Line 66: The authors should clarify what is meant by “neighboring reactions” .

Response: the neighboring reactions refer to the reactions which share common substrates/products and there are existing edges between them in the metabolic network.

Line 69: The citation provided does not support the accompanying statement.

Response: we removed this citation from the revised manuscript.

Line 71: It is not clear what is meant by the statement that metabolic systems are “rather dynamic” in adjusting to perturbations. It seems obvious that introducing a perturbation to a system would result in a dynamic response by the organism. In the case of a genetic perturbation, the strain must be constructed before it is cultured. For this case, the statement is misleading if not entirely incorrect. Did the authors mean ‘robust’ instead of ‘dynamic’ ?

Response: “dynamic” here means that genetic perturbations result in integrated adjustment at the multi-omics level, e.g., the metabolite concentration varying, the flux rewiring and gene expression varying. We changed it to “dynamic in wholistically adjusting” in the revised manuscript.

Line 74-75: The BRENDA database contains a wealth of substrate-level regulatory interactions from reactions across genome-scale metabolism. Substrate-level regulations are particularly well documented for model organisms, such as those modeled in this study. This statement should be modified to reflect this.

Response: we modified this description as “The mechanism of fast-acting and local substrate-level regulation of enzyme activity is well established, including allosteric regulation, feedback inhibition, and covalent modification. Similarly, transcriptional regulation of metabolism on a genome-scale has just begun. However, the expression profiles of metabolic genes are primarily regulated by the global growth state via the sequestration or release of TFs with the variation of concentration of growth indicator metabolites, as shown in the activities of over 200 TFs showing strong correlations with few cognate metabolites following the transition from starvation to growth in *E. coli*.”

Line 75-76: Transcriptional regulation of gene expression is well-documented for model organisms like E. coli for which information is consolidated in the regulonDB database. This statement should be updated to reflect this.

Response: We revised the statement to reflect the point raised.

Line 80-82: It is unclear which if the regulatory metabolites referred to are transcription factor effector metabolites, or allosteric effectors of metabolic enzymes.

Response: They are transcription factor effector metabolites and we clarified this in the revised manuscript as “Particularly the triple regulation relationship of metabolites->TFs->transcription cannot sufficiently reflect the reaction flux variation and this insufficiency has hampered the integration of transcriptional regulation with the current GSM. “

Line 89-93: Knowledge of key metabolites controlling metabolism is not useful without

information about their pool sizes. This statement should be updated to reflect this and the need for metabolomics data in the k-Decrem framework.

Response: A great point. We removed the above statement from the Introduction. Instead, we classify the key metabolites as the biomass metabolites and regulator metabolites, based on their concentrations in the Results section.

Line 101: Is Decrem the name of the model, or the name of the method which is used to constrain the model?

Response: Decrem is the name of the method used to reconstruct the metabolic model.

Line 107-110: It is unclear from this sentence if metabolite concentrations are predicted by the k-Decrem, or are a data requirement for deploying k-Decrem (although mentioned later in Results).

Response: The metabolite concentrations are a data requirement for deploying k-Decrem. In the revised manuscript, we expanded Decrem to have k-Decrem functions.

Line 111: It is unclear what “genome-scale knockout strains” refers to – this sentence needs to be rephrased.

Response: It refers to the single-gene deletion strains of *E. coli* from the Keio collection used in this work: more than 1,000 metabolic gene knockout strains are studied in this paper.

Line 111: It is unclear what is meant by “buffered’ by highly coupled reactions”, particularly if intracellular perturbations are genetic perturbations, as previously stated, a genetic intervention would likely have the same impact on a reaction coupled to the knocked out reaction. This statement should be clarified.

Response: “buffered’ means that when coupled reactions-associated genes are knocked out, then the observed growth rates are much lower than uncoupled reactions.

Line 143-145: It is not clear how single K_m values were determined for coupled and uncoupled reactions. There should be many K_m values, given the multiplicity of coupled reaction sets and the many uncoupled reactions. Statistical quantities (p -value) provided also require an explanation as to how they were implemented and why they were used.

Response: we obtained each reaction’s K_m from the Brenda database, and then classified them into two groups based on whether the associated reactions are coupled or uncoupled. As the K_m values of the reactions don’t follow the normal distribution, we used the Wilcoxon rank test to detect whether the K_m values of coupled reactions have a smaller median value than the uncoupled reactions. And we used the median K_m values of the coupled reaction group and uncoupled reaction group as the K_m values for each group.

Line 145: Any statement about physical proximity is unfounded, as the modeling framework in question does not take into account the spatial organization of cells.

Response: we replaced the “physical proximity” with “biochemical proximity” in the revised manuscript.

Line 145-148: The physiological meaning of K_m is the substrate concentration at which the reaction rate is half of the maximum. This could indicate which reactions are likely operating at or close to V_{max} in a pathway, but there is no evidence that suggests reactions “interact” based on K_m values alone. Neither do cells prioritize metabolic function, nor are pathways capable of “reaching out” to each other.

Response: There are several lines of evidence suggest otherwise. For one, most metabolic fluxes are tightly locally coordinated as enzyme deletions in Keio collection of *E. coli* mutants yielding the largest metabolic changes up to two enzymatic steps distance, while distal connections also exist (Fuhrer *et al.*, 2017). Second, reactions in the same topological modules tend to have similar K_m and K_{cat} values (Li *et al.*, 2019). Third, Nam *et al.* (Nam *et al.*, 2012) find the specialist enzymes (i) are frequently essential; (ii) maintain higher metabolic fluxes; (iii) require more regulation of enzyme activity to control metabolic fluxes in dynamic environments than do generalist enzymes; and (iiii) often have lower K_m values. They also show that cell growth rarely directly depends on flux through generalist enzymes, whereas many specialist reactions are essential for growth across all 174 tested media conditions. Based on these associations, we assume those reactions are coupled to carried out some specific functions; in other words, the metabolic system prefers to utilize some enzymes (reactions) rather than other enzymes over evolution, not all the reactions being equally replaceable.

Line 149: Citation 35, Bordbar *et al.*, does not contain experimentally measured fluxes.

Response: we replaced the paper with “Ishii N, *et al.* Multiple High-Throughput Analyses Monitor the Response of *E. coli* to Perturbations. *Science* 316, 593-597 (2007).” in the revised manuscript.

Line 151-154: It is not clear how uncoupled reaction sets were defined for the comparison in Figure 2C. Were reactions randomly sampled from the set of all uncoupled reactions, normalizing for the size of the coupled reaction sets?

Response: the uncoupled reactions subset used in Figure 2c are randomly selected from the entire set of uncoupled reactions, and the number of uncoupled reactions is the same as that for the corresponding coupled reactions used for comparison in Figure 2C.

Line 168-170: Because the emphasis of this work is on network topology and regulation, particularly in the case of *E. coli*, the authors should see if they can establish a correlation between SLBs and specific operons, this would provide more concrete evidence that their workflow is capturing biology, and not just an artifact of the decoupling framework.

Response: Thank you for this great suggestion. We provided a heatmap to show the correlation between the SLBs and TFs (transcription factors), as we found the operon is a smaller regulation unit than the SLBs (Fig. R9). We can find significant TF enrichment pattern for SLBs, which indicate the functional consistency within the SLBs. Most enriched TFs are part of the general DNA-binding transcriptional regulators, such as SoxS and NrdR. Cell metabolic state (superoxide or nitric oxide, and ATP concentration) often determines these

regulations.

Fig. R9. The heatmap of identified SLBs-related TFs enrichment. The rows are the SLB elements reaction names, and columns are TFs.

Figure 2E: why do low reaction fluxes deviate most in the decomposed vs original model? Do these low fluxes necessarily constitute trivial reactions, as their numerical values are determined by factors such as the coefficient of the end metabolite (of a given pathway) in the sink reaction (such as biomass). Can other basis of comparisons be also reported, such as net ATP/biomass/cofactor yield?

Response: In the initial submission, the trivial reaction means the reactions loaded with a flux less than 1, so the low-flux reactions are the same as the trivial reaction. In the revision, to avoid ambiguity, we renamed the trivial reactions and primary reactions as low-flux reactions and high-flux reactions, as the reviewer mentioned. We checked the low-flux reactions and found that most reactions with deviated fluxes are exchange reactions and located at the end of pathways, e.g., amino acids, Acyl-phosphate, NADH et al. An interesting observation is that almost all those reactions have several parallel sub-reactions, e.g., Nucleoside-diphosphate kinase (ATP:UDP), Nucleoside-diphosphate kinase (ATP:CDP), Nucleoside-diphosphate kinase (ATP:dTDP) and Nucleoside-diphosphate kinase (ATP:dGDP) are reactions catalyzed by the same enzymes with four different cofactors. As the component reactions of each SLB are combined into one, the end metabolites (of a given pathway) will relate to fewer intermediate metabolites and more independent synthetic path, which change the solution space of those end metabolites. The details of flux range distribution across each pathway can be found in Fig. R10.

Line 185: “Decrem” is defined here as a model, but three models are constructed. It needs to be made clear whether “Decrem” is a model or if it is the workflow

forreconstructing/constraining the model.

Response: Thanks for this suggestion! This work is a method paper and Decrem is a workflow for reconstructing and constraining decoupled genome-scale metabolic models by incorporating network topology and local regulation, which approaches metabolic fluxes better than other existing methods. We made this distinction in the revision. Work is underway to collect more multi-omics data to make it a model.

Line 188-189, 194: If the Decrem FVA solution space preserves the original GSM model FVA solution space, then it is unclear what advantage Decrem offers over standard FBA or FVA - does Decrem help reduce the original FVA flux bounds, thereby reducing uncertainty? No citation should be given after this sentence if it is truly a result of the modeling framework that is proposed here. Also (Line 194), FBA predicts a single feasible point within a solution space. To demonstrate improved Decrem predictive capability, the authors should demonstrate how the FVA ranges predicted by Decrem shrink compared to FVA ranges predicted with the original genome-scale model while still capturing the 95% confidence intervals estimated using ¹³C-metabolic flux analysis.

Response: Thank you for pointing out this problem. We corrected the statement of this part in the revised manuscript as “Decrem changes the distribution of flux variability of GSM and makes the distribution of flux variability in central metabolism more consistency with the experimental ¹³C flux distribution, which may reduce the uncertainty of original GSMs”. We then rerun a the (Flux variability analysis) FVA as the reviewer suggested and found that the flux variability of each reaction from Decrem has a very different distribution than the original GSMs (Fig. R10): the flux variability appears to be related to their position in the network topology: the flux variability of central metabolism is expanded (e.g., TCA and PPP) while that of the peripheral pathways is reduced (e.g. membrane lipid metabolism, murein metabolism). When examining the consistency with the experimentally measured flux variability distribution in *E. coli* and *yeast* on aerobic growth, we found more agreement of Decrem with the ¹³C fluxes under 95% of confidence intervals than the original GSMs with FVA (Fig. R11), which shows the potential of uncertainty reduction of Decrem models.

Fig. R10(A). Comparison of the flux variability distribution across each pathway for Decrem and original iML155 and iMM904 model, respectively. The pink dash line indicates the consistency level of compared flux variability range.

Fig. R10(B). Comparison of the flux distribution under 95% confidence interval across non-zero flux reactions predicted from the Decrem (red) and original (purple) iML155 and

iMM904 model with the experimental ^{13}C fluxes (green), respectively.

Line 190: It is unclear what constitutes a “trivial flux”, and what constitutes a “primary flux”. This should be clarified.

Response: We changed “trivial flux” and “primary flux” to “low-flux reactions” and “high-flux reactions”, respectively, for clarity.

Line 193: This statement makes it seem as if Decrem is a single model which is used to predict phenotype for each of the three listed organisms. It should be made clear that the Decrem workflow was used to reconstruct GSMs for the three mentioned organisms.

Response: As suggested, we modified this statement to “we first utilize Decrem to reconstruct metabolic model for three species and then...” in our revised manuscript.

Line 208: To demonstrate activity, it should be demonstrated that the lower bounds for the Decrem-predicted flux distribution is non-zero. To truly demonstrate agreement with ^{13}C MFA-estimated flux, ^{13}C -MFA should be carried out with a genome-scale atom mapping model, and it should be demonstrated that the lower bound of the reactions in question are non-zero when all alternative atom mappings are present in the atom mapping model.

Response: Thanks for pointing out this. We modified the active reactions as the common non-zero flux reactions in the predicted and ^{13}C -MFA measured in the revised manuscript.

Line 210-212: FVA ranges predicted by Decrem should be compared to those predicted by the original stoichiometric model and those estimated with ^{13}C -MFA to demonstrate quantitative agreement between the Decrem-predicted flux ranges and ^{13}C -MFA-estimated flux ranges.

Response: We compared the predicted fluxes from Decrem-reconstructed models with ^{13}C -MFA-estimated flux ranges by FVA, and got a more consistent flux ranges against the flux range from original GSM (Fig. R11). There is a high consistency in flux range between Decrem’s predictions and ^{13}C -MFA fluxes, and the prediction of original GSM is also comparable. We calculated the Jaccard index of each prediction with the ^{13}C -MFA fluxes and obtained the average score for Decrem and original GSM, 0.78 and 0.76, respectively.

Fig. R11. The comparison of flux range predicted by FVA from Decrem (pink), original GSM (purple) vs ¹³C-MFA fluxes range (green) on the *E. coli* ML1515 model.

Figure 3C: why were only TCA cycle reactions selected for the comparison? How are the reported performance metrics affected when the entire network (if experimental fluxes are available), or rest of the core metabolism is considered?

Response: As suggested by the reviewer, we checked the rest of central metabolism and found the identified specific reactions by Decrem in *E. coli* GSMs also agree with the ¹³C-MFA, which include some reactions: PYK (Glycolysis Gluconeogenesis), GLYCL (Folate_Metabolism) NADTRHD (Oxidative_Phosphorylation), and GND (Pentose_Phosphate_Pathway). The common characteristic of those reactions is that they are all highly coupled topologically.

Line 226-227: The authors mention “cellular compartments”, but the cited source for S. cerevisiae 13C-MFA fluxes also does not distinguish between compartments in their core metabolic model. It is, therefore, not clear as to how compartment-wise flux allocation was determined and compared to Decrem predictions.

Response: For mitochondrial reactions, we generated a reference flux distribution for *S. cerevisiae* MM94 model from the wildtype ¹³C-MFA fluxes using ¹³C-flux constrained FBA, compared these reference fluxes with the predicted reaction fluxes from Decrem and original GSMs. The number of typical non-zero fluxes is 76 and 50, and the Spearman correlation coefficients are 0.674 and 0.462, respectively. The specific nonzero fluxes identified by Decrem (vs the original GSMs) can be found in the reactions below:

- '2 Dehydro 3 deoxy D arabino heptonate7 phohosphate mitochondrial transport via diffusion'
- '3 4 hydroxyphenyl pyruvate mitochondrial transport via proton symport'
- 'Acetaldehyde mitochondrial diffusion'
- 'Acetyl CoA hydrolase mitochondrial irreversible'
- 'Acetate transport mitochondrial'
- 'L alanine transaminase mitochondrial'
- 'Alcohol dehydrogenase reverse rxn acetaldehyde ethanol mitochondrial'
- 'Citrate transport mitochondrial'
- 'Carnitine O aceyltransferase mitochondrial'
- '2 deoxy D arabino heptulosonate 7 phosphate synthetase mitochondrial'
- 'D lactate dehydrogenase cytosolicmitochondrial'
- 'L glutamate 5 semialdehyde dehydratase reversible'

mitochondrial' 'Glycine cleavage complex lipoamide mitochondrial' 'Glycine cleavage system lipoamide irreversible mitochondrial' 'Glycine hydroxymethyltransferase reversible mitochondrial' 'Glycine mitochondrial transport via proton symport' 'L Lactate dehydrogenase cytosolic mitochondrial' 'NH₃ mitochondrial transport' 'Proline oxidase NAD mitochondrial' 'L proline transport mitochondrial' 'D1 pyrroline 5 carboxylate dehydrogenase mitochondrial' 'Serine mitochondrial transport via proton symport' 'Succinate transport mitochondrial' 'Succinate fumarate transport mitochondrial' 'Tyrosine transaminase irreversible mitochondrial' 'Tyrosine mitochondrial transport via proton symport', and 'Valine reversible mitochondrial transport via proton symport'.

Line 228-252: It is poor practice to compare GSM-predicted fluxes from a single solution within the solution space to ¹³C-MFA-predicted fluxes. Instead, FVA ranges predicted by each modeling framework should be compared to ¹³C-MFA-estimated confidence intervals. The lack of compartmentalization in yeast atom mapping models also needs to be accounted for when comparing GSM predictions.

Response: we carried out the FVA for the yeast *MM904* model in 38 mutant strains and *E. coli iML1515* model in 28 mutant strains to assess the consistency of predicted flux range by Decrem and original GSMs with the ¹³C-MFA using the Jaccard index, and found a higher consistency between Decrem-predicted and ¹³C-MFA fluxes (Fig. R12A-12C). In comparison with other methods, interestingly, only the mitochondrial reactions fluxes from the Decrem can predict the *ZWF1* strain with an increasing flux in Yeast (Fig. R12D), which is explained as “Inactivation of the oxidative PP pathway branch in the *zwf1* mutant was compensated by a reversed flux in the non-oxidative PP pathway to provide the biomass precursors pentose 5-phosphate and erythrose 4-phosphate. Because the primary role of the PP pathway on glucose is generation of NADPH, NADP⁺-dependent mitochondrial malic enzyme flux was significantly increased in the *zwf1* mutant” in a paper by Sauer group [6].

Fig. R12 (A). Jaccard index metric distribution of predicted fluxes by Decrem vs. the ¹³C-MFA (blue boxplot), as well as the predictions with original GSM vs the ¹³C-MFA (red boxplot) for yeast *MM904* model in 38 mutant strains.

Fig. R12(B). Jaccard index metric distribution of predicted fluxes by Decrem vs. the ^{13}C -MFA (blue boxplot), as well as the predictions with original GSM vs the ^{13}C -MFA (red boxplot) for yeast *MDN750* model in 38 mutant strains.

Fig. R12 (C). Jaccard index metric distribution of predicted fluxes by Decrem vs. the ^{13}C -MFA (blue boxplot), as well as the predictions with original GSM vs the ^{13}C -MFA (red boxplot) for *E.coli* *ML1515* model in 28 mutant strains.

Fig. R12 (D). The comparison of predicted fluxes of mitochondrial reactions among RELATCH, REPPS pFBA, FBA, and Decrem across 38 yeast mutant strains on the *MM904* model.

L228-277: It is not clear how cases with gene knockouts in reactions belonging to a coupled set (hence merged into a linear basis reaction (LBR)) were handled. Was flux through the LBR forced to be zero or were coupled reaction sets recomputed, leaving out the affected reaction? Similarly, can the kinetic parameters estimated for a LBR be interpreted physiologically, perhaps corresponding to an ‘effective’ turnover rate?

Response: In our method, we assign the knockout gene-associated reactions as a close to zero flux (1 in practice), forcing the corresponding LBR to be a small value. Following the reviewer's suggestion, we checked the kinetic parameters for LBRs and did not find direct relationships between them and physiology. The thermodynamics and enzymes should be considered if we would like to interpret the LBR kinetics.

Line 260: The PCA acronym is never defined in the text.

Response: Thanks for pointing this out. We fixed it in the revised manuscript.

Line 264-267: Can the authors provide clarification as to why the maximum growth rate predicted by Decrem is not identical to that predicted by FBA with the original model if both maximize growth rate? Are there growth-coupled reactions which are being removed from the Decrem network?

Response: We carefully checked the reconstructed GSMs and found that the flux range overlapped with the original GSMs (Fig. R13) rather than a subset of the original GSMs. There are different numbers of reactions with different flux ranges between the reconstructed Decrem and original GSMs, which means that the vertex of convex polyhedral constrained by the reaction flux interaction varies with the flux range and that the optimal point of growth rate is often located in the vertex of feasible region-derived convex polyhedral. This is the property of current linear programming.

Fig. R13. Comparison of the flux variability distribution across the pathways for Decrem

reconstructed and original *ML155*. The pink dash line indicates the consistency level of compared flux variability range.

Line 267-269: Did the other methods predict similar growth rates for the mentioned strains?

Response: Similar growth rates have can in this study. However, as shown in Fig. R14, we can see that only the mitochondrial reactions fluxes from the Decrem are able to predict the ZWF1 mutant strain with an increasing flux, which is consistent with the “Inactivation of the oxidative PP pathway branch in the *zwf1* mutant was compensated by a reversed flux in the non-oxidative PP pathway to provide the biomass precursors pentose 5-phosphate and erythrose 4-phosphate. Because the primary role of the PP pathway on glucose is generation of NADPH, NADP⁺- dependent mitochondrial enzyme flux was significantly increased in the *zwf1* mutant” in a paper by Sauer group [6]. Among the methods compared, only RELATCH predicts the mitochondrion-related reactions as a decreasing flux, while the rest predict it as insignificant.

Fig. R14. The comparison of predicted fluxes of mitochondrial reactions by RELATCH, REPPS, pFBA, FBA, and Decrem across 38 yeast mutant strains on the *MM904* model.

Line 267-269: If Decrem was not used to predict the genetic mutations leading to low growth rate, then please rephrase to state that Decrem was able to predict a low growth rate in those strains, rather than identify the strains.

Response: Thanks for this suggestion. This part was rephrased in the revised manuscript.

Line 275: “Large-effect mutations” is un-descriptive, and the authors make the claim that other constraint-based methods lead to a high false-positive rate when assessing growth rate. The authors should clarify what is meant by “false-positives”, and provide specific examples where Decrem is able to better predict growth phenotypes than other constraint based methods (the ZWF1 example in the text fails to discuss predictions made by the other methods).

Response: Here, the “false-positives” means the ZWF1 is predicted as an abnormal flux distribution by RELATCH, REPPS, and FBA, but the mitochondrial fluxes predicted by these three methods are either not significant or decreased against the wildtype, only Decrem gets a consistency prediction with the experimental result. We removed the “false positives” in the revised manuscript.

Line 290-291: Particularly for E. coli, for which transcriptional regulatory information is consolidated in a single database, there is a wealth of information linking effector metabolites with transcription factors and transcription factors with regulated enzymes. It should be made clear how the developed framework both supports the existence of these documented effector metabolites and expands on the number of potential effector metabolites.

Response: Thanks for this great suggestion! Based on the paper of Sauer group [3,7], we have here two types of gene expression regulators: local gene regulators, which physically interact with TFs but contribute marginally to gene expression’s interpretation (regulation), and global cell state regulators which contribute predominantly to the interpretation. Furthermore, we mainly focused on the global regulator-dominated genes in this work (see the detailed response to the following comment); they often do not interact with TFs directly. Furthermore, several recent studies have proved the existence of global state regulators and indicated that amino acid (global regulators) degradation metabolites pyruvate and oxaloacetate could directly inhibit the phosphotransferase system (PTS), which means the global TF Crp-derived indirect regulatory mechanism [7-8].

Lines 289-356: No mechanistic basis for the transcription model developed in this study is presented. The Michaelis-Menten formalism describes allosteric interactions with the enzyme itself. The authors should provide a mechanistic meaning for the developed model which links the kinetics of transcription to enzyme kinetics if the intent is to describe transcriptional regulation using enzyme kinetics.

Response: We developed a mechanistic basis model to link transcription kinetics to enzyme kinetics based on the gene regulatory model from Sauer group [3]. From Sauer’s model, we can get a linear regulation model between the central metabolism gene activity and local and global regulators.

$$\log(E_g) \approx \alpha_g \log(R) + \sum_{i=1}^K \beta_{gi} \log(M_{gi})$$

where the E_g represents the expression E of gene g , R indicates the given growth rate, and M_{gi} represents the i th of K metabolite regulators, then α_g and β_{gi} represent the corresponding coefficient. According to the biomass reaction, we can represent the growth rate follows:

$$R = \lambda \prod_{j=1}^N (1 + M_{bj}/K_{mj})^{\theta_j}$$

where the M_{bj} represents the j th of N biomass metabolites, and K_{mj} is a cell state-related kinetic parameter, λ and θ_j are the reaction coefficients, following the Sauer paper, we approximate $\log(1 + M_{bj}/K_{mj})$ with $\log(M_{bj}/K_{mj})$, then we take logarithm operation for

above equation and approximate it as:

$$\begin{aligned}\log(R) &= \sum_{j=1}^N \theta_j \log(1 + M_{bj}/K_{mj}) + \log(\lambda) \\ &\approx \sum_{j=1}^N \theta_j \log\left(\frac{M_{bj}}{K_{mj}}\right) + \log(\lambda) \\ &= \sum_{j=1}^N \theta_j \log(M_{bj}) + c\end{aligned}$$

So

$$\log(E_g) \approx \alpha_g \sum_{j=1}^N \theta_j \log(M_{bj}) + \sum_{i=1}^K \beta_{gi} \log(M_{gi}) + b$$

where M_{bj} indicates the biomass metabolites and M_{gi} represents the TF regulatory metabolites, we get an approximate quantitative relationship between gene expression and metabolite concentration in the central metabolism.

We then validated the above linear relationship between the normalized metabolic regulators and gene expression in the central metabolism of *E. coli*. We first tested the correlation score between 85 genes expression and concentrations of 45 identified potential regulator metabolites (which consist of the regulators in Sauer's paper [3] and biomass composition [7]) in the central metabolism of *E. coli* on 24 mutant strains using the CCA (Fig. R15A). However, we got a poor correlation score across their top two canonical variables (representing 90% variance):

Next, we clustered the correlation between the concentrations of 45 metabolites and 85 genes expression, and obtained two groups: the biomass group and the regulator group (Fig. R15B). We then tested the correlation between the concentration of metabolites in either group with the 85 gene expressions and found that all canonical variables show correlation coefficients greater than 0.8 in the biomass group, whereas no significant correlations were observed in the regulator group in the regulator group (only first canonical variable correlates 0.5; Fig. R15C and R15D). These observations are consistent with the established knowledge: central metabolism genes are primarily regulated by the general global cell state and enhanced by specific regulators [3,7-8]. This result explains the linear relationship between biomass composition and gene expression in central metabolism, and we constructed a regression model to quantify it.

Considering the complex and specific local regulators for TFs, in this paper, we only sought to identify the genes dominantly regulated by the global cell state. We used a PLSR to fit the 85 gene expressions with biomass group metabolites with leave-one-out cross-validation, and filtered the genes with a cutoff regression coefficient of 0.84. The first PC 0.38 [accounted for 38% of variance] as the global cell state-derived genes, 32 genes are selected from the total of 85 genes in central metabolism, and most of which are in PPP pathway, pyruvate metabolism, and oxidative phosphorylation. We then calculated the correlation between the identified genes and biomass metabolites using CCA and got a correlation coefficient greater than 0.9 for each canonical variable (Fig. R15E).

Fig. R15A. The CCA correlation between the 45 metabolite concentrations and 85 gene levels in central metabolism of *E. coli*.

Fig. R15B. Correlation heatmap show the candidate regulators are clustered into two groups: the biomass group and local regulators group.

Fig. R15C. The CCA correlation between the biomass composition metabolite concentration and 85 gene level in central metabolism of *E. coli*.

Fig. R15D. The CCA correlation between the specific regulator metabolite concentration and 85 gene level in central metabolism of *E. coli*.

Fig. R15E. The CCA correlation between the biomass metabolite concentrations and the identified 32 gene levels in central metabolism of *E. coli*.

Fig. R15F. The CCA correlation between the biomass metabolites and identified 32 genes in central metabolism of *E. coli* on a time series dataset.

We then validate identified 32 genes and biomass group metabolites on our experimental dataset (see manuscript for detail) and also find a strong canonical correlation score (Fig. R14F). Finally, to estimate the statistical significance for the selected biomass metabolites, we randomly sampled 10000 times on the 45 metabolites and estimated the regression coefficient for 32 genes with those sampled metabolites. The *P* value is 0.0031 for biomass metabolites

vs. 0.48 for local regulator metabolites, which validates the identified relationship between the gene expression and biomass metabolites is specifically compared to the random sampling.

For the reactions catalyzed by the cell state-dominated gene products, we can reconstruct the Michaelis equation with only metabolite-based formula:

$$\begin{aligned}
 -\log(v([S], [P], [A], [I], [E])) &= \underbrace{-\log([E_g])}_{\text{Enzyme term}} - \underbrace{\log\left(1 - [P]/[S] e^{-\Delta_r G' / RT}\right) + \sum_v \log(1 + [I_v])}_{\text{Metabolite associated terms}} \\
 &= \underbrace{-\log(k^+) - \sum_v k_f^v}_{\text{Kinetic constants}} + \underbrace{\sum_u \log\left(1 + \frac{k_A^u}{[A_u]}\right) + \log\left(1 + \frac{k_m^s}{[S]}\left(1 + \frac{[P]}{k_m^p}\right)\right)}_{\text{Nonlinear terms}}
 \end{aligned}$$

$$\log(E_g) \approx \alpha_g \sum_{j=1}^N \theta_j \log(M_{bj}) + b$$

This formulation can be approximated as a linear kinetic (see the method in revised manuscript):

$$\begin{aligned}
 &\log(v([S], [P], [A], [I], [E])) \\
 &\approx \alpha_g \sum_{j=1}^N \theta_j \log(M_{bj}) + \log(1 - [P]/[S] e^{-\Delta_r G' / RT}) + \frac{[P]/[S]}{k^{eq}} + \sum_u \left(\frac{k_A^u}{[A_u]}\right) \\
 &+ c
 \end{aligned}$$

Lines 336-341: The observation that the expression of growth-associated pathways correlate to the concentration of biomass components is expected. It would be more interesting if the authors could relate gene expression to other effector metabolites that interact with transcription factors such as cAMP, and relate those interactions back to transcriptional regulations documented in regulonDB (in the case of E. coli).

Response: This is a great point! In Sauer's paper, we checked cAMP, ATP, Malate, F6p, G1p, pyruvate and NADP, which are identified as physically interacting with TFs [3]. We tested the correlation with gene expression using the above CCA and PLSR methods with a lower cutoff (first PC has a correlation $R > 0.33$ and total regression coefficient 0.7). Indeed, we identified ten genes: 'zwf', 'rpe', 'aceE', 'ppc', 'frdA', 'frdB', 'frdC', 'frdD', 'acs', and 'ackA'. These genes are a subset of 32 growth-related genes mainly located in the PPP and oxidative phosphorylation. They are explained as the regulation target of global transcriptional factors: Crp and Cra across the paper [3,8].

Line 348-356: If the kinetic model is predicting fluxes for the same conditions that it was fitted to, then the model is not predictive and merely fits the data well. Please clarify whether the model is being used to predict phenotype or if it is only being fitted to data. Details about the kinetic model itself are also absent and should be provided.

Response: In the revised manuscript, the datasets which are utilized to fit and test (predict) our reformulated linear kinetics are independent, e.g., we use multi-omics data with 24 mutant strains from paper [9] as the training dataset, and a dataset with 45-time points from paper [7] as the testing data. By fitting our linear kinetics for identified global-regulation reactions on training set, an outstanding performance is achieved both on training set and testing set (Fig. R16):

Fig. R16. predicted fluxes on training set and test set using the linear kinetic for identified global regulated reactions.

Furthermore, the detailed components of each kinetic reaction are consistent with a linear weight combination of global regulated metabolites and reactant metabolites (Table R2).

Table R2. The components and coefficient of kinetic reactions.

Glucose transport reaction	coefficient	Alcohol dehydrogenase	coefficient	Acetate kinase	coefficient	Pyruvate dehydrogenase	coefficient	Fumarate reductase	coefficient
Glucose + PEP -> G6P + Pyr	0.307186346	Acetate	0.721504165	Ethanol	-0.265069986	Phosphoenolpyruvate		Fumarate	
Glucose + PEP -> G6P + Pyr	0.335726	AcCoA	0.2727	AcCoA	0.335725576	Pyruvate		Succinate	2.074623592
G6P/Glucose	-0.2447958	pyruvate	-0.069537413	pyruvate	-0.307186346	Succinate	3.393716814	Adenosylmethionine	-1.2014664
G6P	-0.244304666	Succinate	6.39336429	Succinate	1.94408948	Adenosylmethionine	-1.676982036	ATP	0.171462394
Succinate	9.386932322	Adenosylmethionine	-13.17472976	Adenosylmethionine	-12.49446116	ATP	1.673871667	Asp	0.831461294
Adenosylmethionine	-2.235293326	ATP	9.41281548	ATP	8.72278255	Asp	3.666045998	Lys	-0.395996955
ATP	4.06264328	Asp	9.49177552	Asp	4.6812473	Lys	-1.299327623	Adenosine	-0.539635646
Asp	2.056865279	Lys	3.82454359	Lys	7.9399822	Adenosine	-0.140792936	Tyr	31.07717952
Lys	-1.10958472	Adenosine	3.8026796	Adenosine	1.61093147	Tyr	41.81461614	Gly	-2.948042664
Adenosine	-1.452019351	Tyr	1.91651857	Tyr	2.13382173	Gly	-13.40582131	1,4-Butanediamine	-0.803296926
Tyr	154.0302055	Gly	2.88115423	Gly	1.30264384	1,4-Butanediamine	-6.654853931	Ala	-2.095757571
Gly	-6.907092627	1,4-Butanediamine	13.46265733	1,4-Butanediamine	1.89127443	Ala	-5.530132079	Ser	5.409127724
1,4-Butanediamine	-1.087397455	Ala	-9.46658315	Ala	-6.88765129	Ser	13.9701257	Val	0.15580263
Ala	-4.657589052	Ser	0.06755964	Ser	1.14228142	Val	8.207702762	2,4-Diaminobutyrate	-0.16950912
Ser	12.81172352	Val	-13.40624955	Val	-7.82691485	2,4-Diaminobutyrate	-0.159443808	Thr	0.299317114
Val	-11.00880606	2,4-Diaminobutyrate	-0.05134267	2,4-Diaminobutyrate	4.1689483	Thr	-1.890708431	Asn	9.703027946
2,4-Diaminobutyrate	-1.001734988	Thr	19.65628798	Thr	16.53299959	Asn	30.59856192	AMP	10.0881335
Thr	-7.984767058	Asn	-13.3901532	Asn	-9.2515847	AMP	28.48000738	Adenine	5.057259529
Asn	24.99922773	AMP	17.01019164	AMP	11.48016019	Adenine	8.890065328	Hypoxanthine	1.627991655
AMP	37.59856818	Adenine	-6.88857491	Adenine	-3.74056073	Hypoxanthine	-2.041054333	Phe	-5.784687837
Adenine	6.005114841	Hypoxanthine	0.15615138	Hypoxanthine	2.11667175	Phe	3.078150064	Met	-6.953741309
Hypoxanthine	-7.74331655	Phe	20.02118004	Phe	14.10642532	Met	-1.90681375	Guanine	5.070878738
Phe	5.43148997	Met	-4.52838924	Met	-5.07779626	Guanine	1.139770908	His	-3.925657859
Met	3.84761775	Guanine	-3.44498826	Guanine	-3.41561285	His	0.035929914	Inosine	-2.44786846
Guanine	3.75641404	His	0.49626659	His	-5.312897325	Inosine	0.335725576		
His	2.76192234	Inosine	15.91548003	Inosine	11.33601615		-0.307186346		
Inosine	11.0396602								

Line 361-380: It is presumable the added kinetic constraints caused a decrease in the growth rate of the knockout strains. Can the authors confirm that the improved correlation was due to a decrease in the predicted growth rates?

Response: This is not the case, as the prediction by Decrem with kinetic constraints has a

larger growth rate than the canonical Decrem, which can be found in Fig. R17.

Fig. R17. The predicted growth rate distribution between canonical Decrem and kinetic Decrem on the 1080 *E. coli* mutant dataset.

Line 382: Please define the term “topological vulnerability”.

Response: Thanks for pointing this out. We defined it as “the coupled reactions with topological hub property” in the revised manuscript.

Line 395-399: A discussion of the Relatch and REPPS methods is also warranted, as their predictive accuracy is the same as Decrem and k-Decrem, if not higher.

Response: Thanks for this suggestion. We discussed about the Relatch and REPPS methods in the revised manuscript: “both Relatch and REPPS also produced a comparable accuracy by constraining the identified kinetic fluxes. We estimated the non-zero fluxes distributions of Relatch and REPPS and found that REPPS has a better fluxes distribution than RELATCH compared with Decrem across the pathways (Figure 8B), which may be due to the unavailable enzyme expression data during the RELATCH analysis.”

Figure 8: A comparison with kinetic models of *E. coli* metabolism (such as Khodayari and Maranas, 2016, Nat. Comm., Millard et al., 2017, PLoS Comp Biol., Kurata et al., 2018, J Biosci Bioeng., Gopalakrishnan et al., 2020, Metab Eng.) is also necessary, especially as Decrem is an optimization-based framework and uses biomass as the objective function, which has been shown to not be valid under every growth conditions.

Response: Thanks for this suggestion. First, our identified kinetic reactions are metabolite concentration-based models, which use an approximated way to reduce the complexity of the entire kinetic model. Importantly, there are only five global-regulated kinetic reactions that are used to constrain the corresponding flux distribution. In contrast, the methods mentioned above are multi-omics based model, and predict the genome-scale knockout strains with only the experimental metabolite concentration data. We compared kinetic Decrem with k-Ecoli475 [10] (Khodayari and Maranas, 2016, Nat. Comm), kurata [11] (Millard et al., 2017 and 2018, J Biosci Bioeng. The latter is an updated version of the former method), and k-FIT [12] (2020, Metab Eng.) (We did not include the PLoS Comp Biol., Kurata et al., 2018 as this model is used only for flux control analysis). On an *E. coli* multi-omics multi-knockout strains dataset, we got the following results in Table R3.

Table R3. Comparison of predicted flux consistency with ¹³C fluxes by four test methods.

Strain	kurata	k-ecoli457	k-FIT	k-Decrem
glk	0.76	0.92	0.99	0.89
pgi	0.77	0.93	0.99	0.91
pfka	0.75	0.96	0.99	0.87
pfkb	0.8	0.96	0.97	0.9
fbp	0.83	0.95	0.99	0.89
fbab	0.82	0.98	0.98	0.91
gpma	-0.22	0.99	0.99	0.87
pyka	0.83	0.98	0.98	0.91
pykf	0.89	0.94	0.95	0.87
ppsa	0.78	0.99	0.93	0.89
zwf	0.9	0.95	0.99	0.91
pgl	0.2	0.99	0.98	0.85
gnd	0.24	0.97	0.99	0.89
rpe	0.16	0.99	0.97	0.9
rpia	0.83	0.94	0.95	0.9
rpib	0.8	0.95	0.98	0.91
tkta	0.72	0.95	0.96	0.89
tktb	0.74	0.99	0.99	0.89
tala	0.76	0.99	0.99	0.92
talb	0.76	0.99	0.98	0.9
ppc	0.79	0.93	0.99	0.89
wt 0.1	0.95	0.96	0.97	0.93
wt 0.4	0.94	0.81	0.98	0.88
wt 0.5	0.9	0.95	0.97	0.92
wt 0.7	0.86	0.97	0.98	0.9

Line 453-456: Is Decrem a model or a modeling framework? Throughout the manuscript, it is compared alongside formalisms such as FBA, pFBA, RELATCH, etc. This needs to be made clear.

Response: Decrem is a method framework used to reconstruct and constrain GSMs. We make this clear in the revised manuscript.

Line 469-470: The authors should remove “in biochemistry textbooks” and provide citations to support their statement. If a citation cannot be found to support the statement, then it should be removed. Also, the authors are recommended to use regulonDB for obtaining genome-wide transcriptional regulatory information in E. coli. Please update the statement to reflect this knowledgebase or remove it.

Response: We removed this statement in the revised manuscript.

Line 505: It is not clear what is meant by “coactivated” reaction cycle. The TCA cycle

is given as an example of a “coupled, coactivated reaction cycle”.

Response: We replaced this statement with the “topologically coupled reaction cycle” in the revision.

Line 525, 528, 530: How the equations are used in the context of the Decrem workflow is never stated. This should be made clear in the methods section, perhaps by including the workflow chart given in Supplementary Figure 1.

Response: Thanks for this suggestion. We made it clear there are more than one topologically coupled subnetwork in the GSMs [13]. To explore all those substructures, we first calculated the topological similarity of each pair of reactions in a GSM to form a similarity matrix, then carried out a clustering algorithm to split each subnetwork, and finally, each subnetwork was reconstructed as the combination of LSBs. Following the reviewer’s suggestion, we clarified this point in the revised manuscript and added it to Supplementary Figure 1.

Line 534: Throughout the manuscript, “highly coupled” reactions and substructures are mentioned, but what defines a “highly coupled” set of reactions is never fully defined. The authors should define what constitutes a “highly coupled” reaction.

Response: Thank you for this suggestion. We defined the highly coupled reactions as those within the same substructure identified above.

Line 553: The formulation needs to be restated to be more easily understood. There is also no way to tell what the purpose of Step 2 is in the broader workflow without referring to the supplementary materials. If the authors directly applied the MinSpan algorithm in this step, they should state this with the citation and avoid math notation. If the authors somehow adapted the MinSpan algorithm for a novel use, they should state this and how their formulation differs from the original algorithm.

Response: Thanks for this suggestion. There are two main differences between the MinSpan and our model, (1) the MinSpan was applied to decouple the whole GSMs, while our model is utilized to decouple each identified substructure/network, so we need to introduce some auxiliary exchange reactions to maintain metabolite flux balance;(2) the sparse linear basis vector from MinSpan don’t always guarantee the orthogonality by mixed-integer linear programming as the solution space of GSMs is constrained into a convex cone, our model add an orthogonal constraint with a linear programming when solving the sparse linear basis. Following this suggestion, we add a description to explain this part in the revised manuscript.

Line 576: Can the authors elaborate on what the purpose of this optimization formulation is in the context of the overall Decrem workflow?

Response: This formulation is an FBA-like optimization formulation for the reconstruction GSMs in Decrem. The main differences include: (1) $S_{C_k}^{IBR} = S_{C_k}^* \cdot N_{C_k}^{S^*}$ means each sparse linear basis is combined into one cooperating “reaction” which maintains stoichiometric balance for C_k , the specific subnetwork (cluster). So we here use the coefficient of sparse

linear basis to weight each coupled reaction to reconstruct each cooperating “linear basis reactions”; (2) reassigning the flux bounds for each “linear basis reaction” with $\max\left(f\left(lb_{C_k}^i, NZ(N_{C_k}^i)\right) ./ f_N\right) \leq v_{C_k}^i \leq \min\left(f\left(ub_{C_k}^i, NZ(N_{C_k}^i)\right) ./ f_N\right)$, which means the lower bound of “linear basis reaction” take the maximal value of its element reactions, and the upper bound takes the corresponding minimal value, by $NZ(N_{C_k}^i)$, the normalization of each element reaction bound with corresponding composition coefficient of sparse linear basis.

Line 603: Can the authors provide a mechanistic basis for the expressions used to establish transcriptional relationships in this section? Methods such as that used by Kochanowski et al. to identify regulatory metabolites coordinating gene expression have attached biophysical meaning to the expressions they have used to probe for effector metabolites.

Response: We developed a mechanistic basis model to link the kinetics of transcription to enzyme kinetics based on the gene regulatory model from Kochanowski et al. [5]. From Kochanowski’s model, we can get a linear regulation model between the central metabolism gene activity and local and global regulators.

$$\log(E_g) \approx \alpha_g \log(R) + \sum_{i=1}^K \beta_{gi} \log(M_{gi})$$

where the E_g represents the expression E of gene g , R indicates the given growth rate, and M_{gi} represents the i th of K metabolite regulators, then α_g and β_{gi} represent the corresponding coefficients. According to the biomass reactions, we can represent the growth rate as follows:

$$R = \lambda \prod_{j=1}^N (1 + M_{bj}/K_m)^{\theta_j}$$

where the M_{bj} represents the j th of N biomass metabolites, and K_m is a cell state-related kinetic parameter, λ and θ_j are the reaction coefficients, following Sauer Group’s paper [3], we approximate $\log(1 + M_{bj}/K_m)$ with $\log(M_{bj}/K_m)$. We take logarithm operation for above equation and approximate it as:

$$\begin{aligned} \log(R) &= \sum_{j=1}^N \theta_j \log(1 + M_{bj}/K_m) + \log(\lambda) \\ &\approx \sum_{j=1}^N \theta_j \log\left(\frac{M_{bj}}{K_m}\right) + \log(\lambda) \\ &= \sum_{j=1}^N \theta_j \log(M_{bj}) + c \end{aligned}$$

So,

$$\log(E_g) \approx \alpha_g \sum_{j=1}^N \theta_j \log(M_{bj}) + \sum_{i=1}^K \beta_{gi} \log(M_{gi}) + b$$

where M_{bj} indicates the biomass metabolites and M_{gi} represents the TF regulating metabolites, we got an approximate quantitative relationship between gene expression and metabolite concentration in the central metabolism.

We then validated the above linear relationship between the normalized metabolic regulators and gene expression in the central metabolism of *E. coli*. First, we tested the correlation between 85 gene expression and 45 identified potential regulatory metabolites (which consist of the regulators in Sauer Group's paper [3] and biomass composition) concentration in the central metabolism of *E. coli* on 24 single gene-deletion mutant strains using the CCA (canonical correlation analysis), and got a poor correlation score across their top two canonical variables (representing 90% variance):

Fig. R18A. The CCA correlation between the 45 metabolites and 85 genes in central metabolism of *E. coli*.

Next, we clustered the concentrations of 45 metabolites and obtained two groups, the biomass group and the regulator group. We then tested the correlation between the concentration of metabolites in either each group with the 85 gene expression. We found that all canonical variables show correlation coefficients greater than 0.8 in the biomass group. In contrast, no significant correlations were observed in the regulator group (only first canonical variable has a correlation 0.7, Fig. R18B and R18C). These observations are consistent with the established knowledge: central metabolism genes are primarily regulated by the general global cell state, and enhanced by the specific regulators [3,7]. This result explains the linear relationship between biomass composition and gene expressions in central metabolism, and we constructed a regression model to quantify it.

Considering the complex and specific local regulators for TFs, to make the model simple and constrain the GSMs optimization with primary cell state-regulated reactions in downstream analysis, in this study, we only identify the genes which are primarily determined by the global cell state. We used a partial least squares regression (PLSR) to fit the 85 gene expression with biomass-group metabolites with leave-one-out cross-validation, and filtered the genes with a cutoff regression coefficient of 0.84 and the first PC is large than 0.38 as the

global cell state-derived genes, 32 genes are selected from the total of 85 genes in central metabolism and most of them are in PPP pathway, pyruvate and oxidative phosphorylation (i.e., ATP production). We then calculated the correlation between the identified genes and biomass metabolites using CCA and got a correlation coefficient greater than 0.9 for each canonical variable (Fig. R18D).

Fig. R18B. The CCA correlation between the biomass composition metabolites and 85 genes in central metabolism of *E. coli*.

Fig. R18C. The CCA correlation between the specific regulator metabolites and 85 genes in

central metabolism of *E. coli*.

Fig. R18D. The CCA correlation between the biomass metabolites and identified 32 genes in central metabolism of *E. coli*.

To further validate the linear correlation between the identified 32 genes and biomass metabolites, we performed our experiment using *E. coli* BW25113 and collected its transcriptome and metabolome at four timepoints along the growth curve, each with three replicates in MOPS minimum medium (see Methods). Using these newly generated data, we also estimated the CCA coefficient, and found consistent correlations, as shown in Fig. R18E. Finally, to estimate the statistical significance of the selected biomass metabolites, we performed random sampling 10,000 times for the 45 metabolites and estimated the regression coefficient for 32 genes with those sampled metabolites. The P value is 0.0031 for biomass metabolites vs. 0.48 for regulator metabolites, which validates the identified relationship between the gene expression and biomass metabolites.

Fig. R18E. The CCA correlation between the biomass metabolites and identified 32 genes in central metabolism of *E. coli* on a time series dataset.

Line 603: From reading the methods in the main body of the manuscript, it is unclear how this section ties into the other methods in the workflow. Can the authors link this section to the other sections within the methods section, rather than only doing so in the supplementary methods?

Response: We appreciate this point. To improve the manuscript and its readability, we have almost entirely reorganized and rewritten it. A new schematic diagram for the entire method workflow (Fig. 1 in the revised version) was drawn to explain the connection between this part and the other methods.

Line 686: SIMMER, as described in the accompanying citation, uses convenience kinetic rate laws. Hence, it is unclear why the authors use a logarithmic transformed Michaelis-Menten rate expression for this section.

Response: We remove the SIMMER part from the revised manuscript as local regulators have a little influence on our identified global regulated reactions. (see the coefficient weight in Supplementary Table 6).

Line 686: If SIMMER was used to assess the ability of hypothetical regulations to improve fitness when assembling rate expressions and estimating kinetic parameters, and if regulations not found in literature were included in the kinetic model, then statistical analysis of results for all regulations tested should be included in the manuscript or supplementary files, to provide reasoning for why specific rate expression and parameter values were chosen.

Response: We have removed this part in the revised manuscript.

Line 692-706: This section does not adequately describe how the kinetic formalisms

described in the previous two sections are incorporated into the FBA formulation. The authors should re-write these sections to clearly explain how a steady-state flux distribution is constrained using predictions from the kinetic model and transcriptional model detailed in the previous sections, with attention paid to the manner in which Equation 32 is formulated and described.

Response: We appreciate this suggestion. In fact, this section is very straightforward: we constrain the flux bound of those identified global cell state-regulated reactions with their predicted fluxes from the kinetic model and transcriptional model, it is an independent procedure, just like the extra-flux constraint for the canonical FBA.

Supplementary files:

Materials S3: The authors mention that they have used the transcriptional regulations identified to inform parameterization of a Michaelis-Menten model describing the flux through enzymatic reaction in the metabolic network. Is there a mechanistic basis or historical precedent that allows the authors to express a transcriptional regulation (i.e., data collected from citations 8-10) as an allosteric regulation? Expressing a transcriptional event in this manner is problematic because the mathematical representation of allosteric regulation within the Michaelis-Menten formalism has mechanistic meaning.

Response: Thanks for pointing out this problem. For our identified global regulation-dominated reactions, we limit the allosteric regulators to only ATP, AMP, and Succinate in the reformulated linear kinetic model. In fact, the regression coefficient of reformulated linear kinetic also shows the global-regulation property of identified reactions (Table R4). So the allosteric regulation is limited consideration in our revised manuscript.

Table R4: the linear kinetic of identified global regulation reactions.

Glucose transport reaction	coefficient	Alcohol dehydrogenase	coefficient	Acetate kinase	coefficient	Pyruvate dehydrogenase	coefficient	Fumarate reductase	coefficient
Glucose + PEP -> G6P + PYR	-0.307186346	AcCoA -> AcCoA	0.721504165	Ethanol	-0.265069986	Pyruvate -> Phosphoenolpyruvate	-0.339718814	Adenosylmethionine	-1.2014664
AMP	0.335726	AcCoA	0.2727	AcCoA	0.335725576	Pyruvate	3.393718814	Succinate	2.074623592
G6P/Glucose	-0.2447958	pyruvate	-0.069537413	pyruvate	-0.307186346	Succinate	1.673877667	Asp	0.831461294
G6P	-0.243048652	Succinate	6.39336429	Succinate	1.94408848	Adenosylmethionine	-1.676982036	ATP	0.171462394
Succinate	9.386932322	Adenosylmethionine	-13.17472976	Adenosylmethionine	-12.49446116	ATP	1.673877667	Asp	0.831461294
Adenosylmethionine	-2.235293326	ATP	9.41281548	ATP	8.72278255	Asp	3.666045998	Lys	-0.395996955
ATP	4.062624328	Asp	9.49177552	Asp	4.6812473	Lys	-1.299327623	Adenosine	-0.539635646
Asp	2.058665279	Lys	3.82454369	Lys	7.33959822	Adenosine	-0.140792936	Tyr	31.07717952
Lys	-1.10958472	Adenosine	3.8026796	Adenosine	1.61093147	Tyr	41.81461614	Gly	-2.948042864
Adenosine	-1.452019351	Tyr	1.91651657	Tyr	2.13382173	Gly	-13.40582131	1,4-Butanediamine	-0.803296926
Tyr	154.0302055	Gly	2.88115423	Gly	1.30264384	1,4-Butanediamine	-6.654853931	Ala	-2.095757571
Gly	-8.907092627	1,4-Butanediamine	13.46265733	1,4-Butanediamine	1.89127443	Ala	-5.530132079	Ser	5.409127724
1,4-Butanediamine	-1.087397455	Ala	-9.46058315	Ala	-6.88765129	Ser	13.97071257	Val	0.15590263
Ala	-4.657589052	Ser	0.06755964	Ser	1.14228142	Val	8.207702762	2,4-Diaminobutyrate	-0.16950912
Ser	12.81172352	Val	-13.40624955	Val	-7.82691485	2,4-Diaminobutyrate	-0.159443808	Thr	0.299317114
Val	-11.08980606	2,4-Diaminobutyrate	-0.05134207	2,4-Diaminobutyrate	4.16859483	Thr	-1.890708431	Asn	9.703027946
2,4-Diaminobutyrate	-1.001734988	Thr	19.65628790	Thr	16.53299959	Asn	30.59565102	AMP	10.0861335
Thr	-7.984767058	AMP	-13.3901532	Asn	-9.2515847	AMP	28.48000738	Adenine	5.057259529
Asn	24.9992773	AMP	17.01019164	AMP	11.48016019	Adenine	8.890065328	Hypoxanthine	1.62791655
AMP	37.59856818	Adenine	-6.88557491	Adenine	-3.74056073	Hypoxanthine	-2.041054338	Phe	-5.784987837
Adenine	6.005114841	Hypoxanthine	0.15615138	Hypoxanthine	2.11661715	Phe	3.078150063	Met	-6.953741309
Hypoxanthine	-7.74331655	Phe	20.02116004	Phe	14.10642532	Met	-1.906681375	Guanine	5.078087838
Phe	5.43148997	Met	-4.52838924	Met	-5.07779626	Guanine	1.139770908	His	-3.925657859
Met	3.84761775	Guanine	-3.44498826	Guanine	-3.41561285	His	0.035923914	Inosine	-2.44786846
Guanine	3.75641404	His	0.49628659	His	-5.312897325	Inosine	0.335725576		
His	2.76192234	Inosine	15.91548003	Inosine	11.33601615		-0.307186346		
Inosine	11.0396602								

Minor Concerns

Line 168: “gene level” should be “gene expression level”.

Response: Thanks for pointing out this problem. We have corrected “gene level” with “gene expression level” in the revised manuscript.

Line 206: “Empirical fluxes” should be “Experimental fluxes”

Response: we replaced ‘Empirical fluxes’ with ‘Experimentally measured fluxes’ in the

revised manuscript.

Section title ‘Building a metabolic kinetic model incorporating metabolite - TF regulation’ is misleading, as TFs are not explicitly modeled. Recommend changing it to emphasize that regulatory metabolites are modeled.

Response: In the reversed manuscript, we renamed this section as “Integrating global transcriptional regulation-derived key reaction kinetics into Decrem”.

Line 361: repetition of the phrase ‘growth rates’

Response: Thanks for pointing out this, we rewrite this part as “We apply Decrem constructed above to predict the growth rates of *E. coli* genome-scale single-gene deletion mutants, using a dataset in which the growth rates and the concentrations of over 7,000 metabolites have been experimentally measured” in the revised manuscript.

Line 379: “Empirical” should be “Experimental”

Response: Thanks for pointing out this problem. We have modified “Empirical” with “Experimentally measured” in the revised manuscript.

References

- [1] Kochanowski K , Gerosa L , Brunner S F , et al. Few regulatory metabolites coordinate expression of central metabolic genes in *Escherichia coli*[J]. *Molecular Systems Biology*, 2017, 13(1):903.
- [2] Lewis N E , Hixson K K , Conrad T M , et al. Omic data from evolved *E. coli* are consistent with computed optimal growth from genome-scale models[J]. *Molecular Systems Biology*, 2014, 6.
- [3] Kochanowski K , Gerosa L , Brunner S F , et al. Few regulatory metabolites coordinate expression of central metabolic genes in *Escherichia coli*[J]. *Molecular Systems Biology*, 2017, 13(1):903.
- [4] Sánchez BJ, Zhang C, Nilsson A, Lahtvee PJ, Kerkhoven EJ, Nielsen J. Improving the phenotype predictions of a yeast genome-scale metabolic model by incorporating enzymatic constraints. *Mol Syst Biol.* 2017;13(8):935. Published 2017 Aug 3. doi:10.15252/msb.20167411
- [5] Bordbar A , Nagarajan H , Lewis N E , et al. Minimal metabolic pathway structure is consistent with associated biomolecular interactions[J]. *Molecular Systems Biology*, 2014, 10(7).
- [6] Blank LM, Kuepfer L, Sauer U. Large-scale ¹³C-flux analysis reveals mechanistic principles of metabolic network robustness to null mutations in yeast. *Genome Biology* 6, R49 (2005).
- [7] Zampieri, M., Hörl, M., Hotz, F. et al. Regulatory mechanisms underlying coordination of amino acid and glucose catabolism in *Escherichia coli*. *Nat Commun* 10, 3354 (2019)
- [8] Kochanowski, K., et al. (2021). "Global coordination of metabolic pathways in *Escherichia coli* by active and passive regulation." *Molecular Systems Biology* 17(4): e10064.
- [9] Ishii N , Nakahigashi K , Baba T , et al. Multiple High-Throughput Analyses Monitor the Response of *E. coli* to Perturbations[J]. *Science*, 2007.
- [10] Khodayari A , Maranas C D . A genome-scale *Escherichia coli* kinetic metabolic model k-ecoli457 satisfying flux data for multiple mutant strains[J]. *Nature Communications*, 2016, 7:13806.
- [11] Kurata H , Sugimoto Y . Improved kinetic model of *Escherichia coli* central carbon metabolism in batch and continuous cultures[J]. *Journal of Bioscience and Bioengineering*, 2017, 125(2):e0186440.
- [12] Gopalakrishnan S , Dash S , Maranas C . K-FIT: An accelerated kinetic parameterization algorithm using steady-state fluxomic data[J]. *Metabolic Engineering*, 2020.
- [13] Li G , Cao H , Xu Y . Structural and functional analyses of microbial metabolic networks reveal novel insights into genome-scale metabolic fluxes[J]. *Briefings in Bioinformatics*, 2018(4):4.

FUHRER, T., ZAMPIERI, M., SÉVIN, D. C., SAUER, U. & ZAMBONI, N. (2017). Genomewide landscape of gene–metabolome associations in *Escherichia coli*. *Molecular Systems Biology* **13**(1), 907.

- LI, G., CAO, H. & XU, Y. (2019). Structural and functional analyses of microbial metabolic networks reveal novel insights into genome-scale metabolic fluxes. *Briefings in Bioinformatics* **20**(4), 1590-1603.
- NAM, H., LEWIS, N. E., LERMAN, J. A., LEE, D.-H., CHANG, R. L., KIM, D. & PALSSON, B. O. (2012). Network Context and Selection in the Evolution to Enzyme Specificity. *Science* **337**(6098), 1101-1104.

Reviewers' Comments:

Reviewer #2:

Remarks to the Author:

Reviewer #1 concerns

The authors have somewhat improved the manuscript but my original concerns on novelty and clarity largely remain.

Reviewer #2 concerns

The authors have significantly improved the manuscript and addressed clarification and benchmarking questions.

Reviewer #1 concerns

The authors have somewhat improved the manuscript but my original concerns on novelty and clarity largely remain.

Response: We appreciate your time and effort in reviewing our work. We have carefully revised the manuscript, reorganized the introduction, added the data source, code source, modified the figures, tables and supplementary files to improve the clarity of our paper. The modified details are highlighted in the text. Thank you again for your constructive and insightful review. Your feedback has helped us to improve the quality and clarity of our paper.

Reviewer #2 concerns

The authors have significantly improved the manuscript and addressed clarification and benchmarking questions.

Response: We appreciate your time and effort in reviewing our work. Your feedback has helped us to improve the quality and clarity of our paper.